# Climate change in Bangladesh: Temperature and rainfall climatology of Bangladesh for 1949–2013 and its implication on rice yield

Edris Alam[1,2]*, Al-Ekram Elahee Hridoy[1], Shekh Md. Shajid Hasan Tusher[2], Abu Reza Md. Towfiqul Islam[3,4], Md Kamrul Islam[5]

1 Department of Geography and Environmental Studies, University of Chittagong, Chittagong, Bangladesh, 2 Faculty of Resilience, Rabdan Academy, Abu Dhabi, United Arab Emirates, 3 Department of Disaster Management, Begum Rokeya University, Rangpur, Bangladesh, 4 Department of Development Studies, Daffodil International University, Dhaka, Bangladesh, 5 Department of Civil and Environmental Engineering, College of Engineering, King Faisal University, Al Hofuf, Saudi Arabia

* ealam@ra.ac.ae

**Data Availability Statement:** All relevant data are within the paper.

**Funding:** The authors received no specific funding for this work.

## Abstract

Bangladesh has been ranked as one of the world's top countries affected by climate change, particularly in terms of agricultural crop sector. The purpose of this study is to identify spatial and temporal changes and trends in long-term climate at local and national scales, as well as their implications for rice yield. In this study, Modified Mann-Kendall and Sen's slope tests were used to detect significant trends and the magnitude of changes in temperature and rainfall. The temperature and rainfall data observed and recorded at 35 meteorological stations in Bangladesh over 65-years in the time span between the years 1949 and 2013 have been used to detect these changes and trends of variation. The results show that mean annual $T_{mean}$, $T_{min}$, and $T_{max}$ have increased significantly by 0.13˚C, 0.13˚C, and 0.13˚C/decade, respectively. The most significant increasing trend in seasonal temperatures for the respective $T_{mean}$, $T_{min}$, and $T_{max}$ was 0.18˚C per decade (post-monsoon), 0.18˚C/decade (winter), and 0.23˚C/decade (post-monsoon), respectively. Furthermore, the mean annual and pre-monsoon rainfall showed a significant increasing trend at a rate of 4.20 mm and 1.35 mm/year, respectively. This paper also evaluates climate variability impacts on three major rice crops, Aus, Aman, and Boro during 1970–2013. The results suggest that crop yield variability can be explained by climate variability during *Aus*, *Aman*, and *Boro* seasons by 33, 25, and 16%, respectively. Maximum temperature significantly affected the Aus and Aman crop yield, whereas rainfall significantly affected all rice crops' yield. This study sheds light on sustainable agriculture in the context of climate change, which all relevant authorities should investigate in order to examine climate-resilient, high-yield crop cultivation.

## Introduction

It is undeniable that the global climate is changing, and the changes are occurring at both the global and regional scales due to global climate warming [1–4]. (The global average surface

**Competing interests:** The authors have declared that no competing interests exist.

temperature has been projected to increase by 1.4–5.8˚C between the years 1990 and 2100 [5]. For most scenarios, the global surface temperature will likely increase to 1.5˚C and exceed 2.0˚C in several scenarios by the end of the 21st century [5]. There is strong evidence that rainfall pattern variations have occurred on regional [4] and global [6] scales due to global warming. Therefore, understanding climatic trends and long-term changes on the regional and local scales is essential for sustainable agronomic strategies, climate change adaptation, and disaster management planning in Bangladesh.

Bangladesh is projected to experience an increase in average daily temperatures of 1.0˚C by 2030 and 1.4˚C by 2050, indicating the impact of anthropogenic climate change [7]. The rainfall frequency is expected to increase in Bangladesh as well as a global scale [8]. Because regional and local factors influence climate on a local scale, it frequently differs from climate on a global scale [9,10]. Bangladesh is an agricultural country whose economy is firmly based on agriculture, which is vulnerable to climate variability and change. The changing climate leads to natural hazards including floods, droughts, cyclones, tidal surges, and soil salinities [11–14], which threaten agriculture production and food security. Furthermore, floods contribute to the further salinization of coastal lands, which leads to soil salinization and the loss of harvests and productive agricultural land [15]. Climate variability has a severe impact on crop yield variability [16–22]. Thus, the effects of climate change on crop yield are significant concerns for Bangladesh's economic development as well as food security.

Various seasonal climatic factors, such as variations in day and night temperatures and shifting rainfall patterns, significantly impact Bangladesh's agriculture [23–24]. Several studies have been performed to explore climate-yield relationship in Bangladesh at the local level. For example, one study [25] used a linear regression model to examine current and future trends in climatic parameters at the local level and the potential impacts of climate change on paddy production in Satkhira, Bangladesh. Another study [26] analyzed local-level data for climate change impacts on crop yields. A study [27] explored the variability of climatic parameters with three popular rice varieties in northwestern Bangladesh. Recently, another study [22] assessed the trends in climatic factors and their effects on the yield of rice crops in the southern region of Bangladesh. But the earlier studies did not examine the trends in climatic variables at the national level, and regional-level data on yield and weather might not give a clear picture of the relationship between climate and yield because climatic variables affect agricultural production in Bangladesh in different agroecological zones [28]. Hence, the region-specific climate-yield study will help researchers better understand climate change's impact on crop production.

Since non-parametric tests such as Modified Mann-Kendall (MMK) and Sen's slope estimator are not affected by outliers, they are widely used for analyzing trends in temperature and rainfall data [29–32]. The trend analysis of temperature, particularly that of minimum and maximum temperatures monthly and rainfall on a monthly and seasonal basis, both at the national and regional scales, is vital to enhancing the resilience of agricultural production systems to climate change in Bangladesh. The integrating the modified Mann–Kendall test, the Theil-Sen slope method, and backward elimination will provide a better insight into the trend and climatic influence on rice production. On the other hand, parametric test like linear regression will help to detect the trends in timeseries datasets, for example, a study [33] used parametric linear regression to analyze temperature and rainfall trends to detect annual, monthly, and seasonal average maximum, minimum temperature, and rainfall trends from 1976 to 2008. Several studies, such as [24,27,34,35], also tried to find linear trends in the change in temperature and rainfall in Bangladesh. Although numerous studies on the trend analysis of temperature and rainfall data in Bangladesh have been performed in recent decades [21–22,27,35,36], few comprehensive studies on climatic trends regarding rice yield under

climate change have been evaluated [26,37,38]. Due to significant regional variability in temperature [18,23] and irregular rainfall distribution, it is essential to conduct thorough studies at the local and national levels [35].

The IPCC report [39] highlights the necessity for regional or local-level evidence on climatic change. Such studies can aid in meeting policymakers' evidence requirements at the regional level and for various socioeconomic groups. There is a shortage of quantitative rice yield datasets on climate change effects, principally at regional and national levels [20]. This gap occurs in the dataset concerning Bangladesh. This allows for examining whether the climate-yield relationship is a function of a climatic variable at a regional or national level, which is unclear and not sufficiently addressed in the earlier literature. Given this drawback, our research intends to provide insight into climatic trends for rice yield, focusing on climatic change and how such change affects rice production in diverse cropping seasons and agroecological zones.

For this reason, this study has analyzed temperature and rainfall data from 35 stations in Bangladesh using modified Mann-Kendall and Sen's methods. The data from the 35 stations were aggregated to determine whether there were positive, negative, or no significant trends and changes on monthly, seasonal, and annual timescales. This study aims to characterize long-term spatiotemporal climate changes and trends in Bangladesh at the national and regional scales and their implication on rice yield. Seventeen stations, evenly distributed across the country, were selected to understand the spatiotemporal temperature and rainfall trends and changes. In addition, detrended crop yield and climate variables were adopted to realize the potential effect of climate variability on rice yield in Bangladesh from 1970 to 2013. Apart from climate variables and their effects, the present study did not investigate the impact of other relevant factors, like irrigation, crop variety, soil management, use of fertilizer, or pesticides. This research will assist in developing proper agricultural adaptive policies to offset rice yield losses triggered by climate change.

## Study area, data sources and methods

### Study area

Bangladesh spatially ranges from 20° 34′ N to 26° 38′ N latitude to 88° 01′ E to 92° 41′ E longitude. Bangladesh has a humid subtropical climate with wide seasonal rainfall variations, moderately warm temperatures, and high humidity. Bangladesh is located in the tropical monsoon region, and high temperatures, heavy rainfall, high humidity with moderately defined seasonal variations characterize its climate [40,41]. Four distinct seasons can be recognised in Bangladesh:

i. winter from December to February,

ii. pre-monsoon from March to May,

iii. monsoon or rainy season that lasts from June through September and

iv. the post-monsoon season from October to November [42] (Shahid 2009).

There is a variation of temperature in the east and south, which is 27°C to 31°C in the west-central part. The mean temperature from 28°C in the western part of the country to 26°C in the east [42]. The maximum temperature is observed as 40°C in the western regions, and May is the hottest month [43] (Shahid, Harun et al. 2012).IPCC [44] stated that from 1985–1998, Bangladesh has experienced an increasing trend in the average temperature of about 1°C in May and 0.5°C in November. In Bangladesh, the mean precipitation ranges from 1400 mm in the western region to 4400 mm in the eastern region [42].

## Data sources

For this study, the monthly maximum, minimum, and mean temperatures, as well as the rainfall records of 35 stations in Bangladesh across 65 years (1949–2013), were collected from the Bangladesh Meteorological Department (BMD). Of these 35 stations, 65 years of records are not available for Rangamati, Sylhet, Rajshahi, Dhaka, and Rangpur; for these stations, 57 (1957–2013), 58 (1956–2013), 50 (1964–2013), 61 (1953–2013), and 60 (1954–2013) years were available, respectively. However, we used all the available period data from 35 weather stations for the national scale. Because some stations were established after 1990, no long-term data are available at those locations. The meteorological stations were selected based on their location, data availability, and homogeneity, allowing these data to cover all of Bangladesh. This study attempts to cover long-term datasets, which is why the data duration is 1954–2013.

A total of 17 selected stations have more than 49 years of complete records for the period from 1949 to 2008. The total missing data was less than 2%, and missing data was imputed by using the average value of the same month but getting the average of previous and subsequent years and putting values accordingly [9]. While pre-processing the datasets, outliers and redundant data (e.g., negative rainfall values, Tmin > Tmax) were also checked for verification purposes. In addition, we considered all the available data from 35 stations on a national scale. The selected stations are shown in Fig 1. The stations are distributed evenly across the country and are expected to give a whole-country picture. The name, latitude, longitude, and data period used for this study of these stations are provided in Table 1. The available rice yield (t/ha) data for 44 years (1970–2013) was obtained from the Bangladesh Bureau of Statistics (BBS). The three major rice crops are Boro, Aus, and Aman. These growing seasons nearly coincide with three climatic seasons: winter (November to February), hot summer (March to May), and monsoon (June to October) [45].

## Methods

### Homogeneity test

Climatic data often contains inhomogeneity for various reasons. Inhomogeneity within the data may lead to erroneous results. Changes in observing methods, equipment changes, and station relocations can impact trend analysis [46]. In this regard, the Standard Normal Homogeneity Test (SNHT) [47] and Buis-hand's Range (BHR) [48] were used to test the temperature and rainfall time series of each station to detect inhomogeneity in the time series. The test was applied to all stations at a 5% significance level. Generally, the results of the test revealed that all data sets were homogeneous.

### Serial correlation

Serial correlation, also known as autocorrelation, is the relationship between a given variable and itself over various intervals of time. Autocorrelation in time series data is continuously seen as one of the main issues in the analysis and detection of time series trends. It leads to possible incorrect trend significance assessments in the Mann-Kendall test. Having a positive autocorrelation in the data increases the chances that trends would be detected when none exist and vice versa [49]. The Mann-Kendall test, when applied to auto-correlated time series data, gives wrong or too high rejection rates [50].

The auto-correlation coefficient is calculated at lag-k using [51] equation to check the existence of a serial correlation as:

$$r_k = \frac{\frac{1}{n-k}\sum_{t=1}^{n-k}(x_t - \bar{x})(x_{t+k} - \bar{x})}{\frac{1}{n-1}\sum_{t=1}^{n-k}(x_t - \bar{x})^2}$$

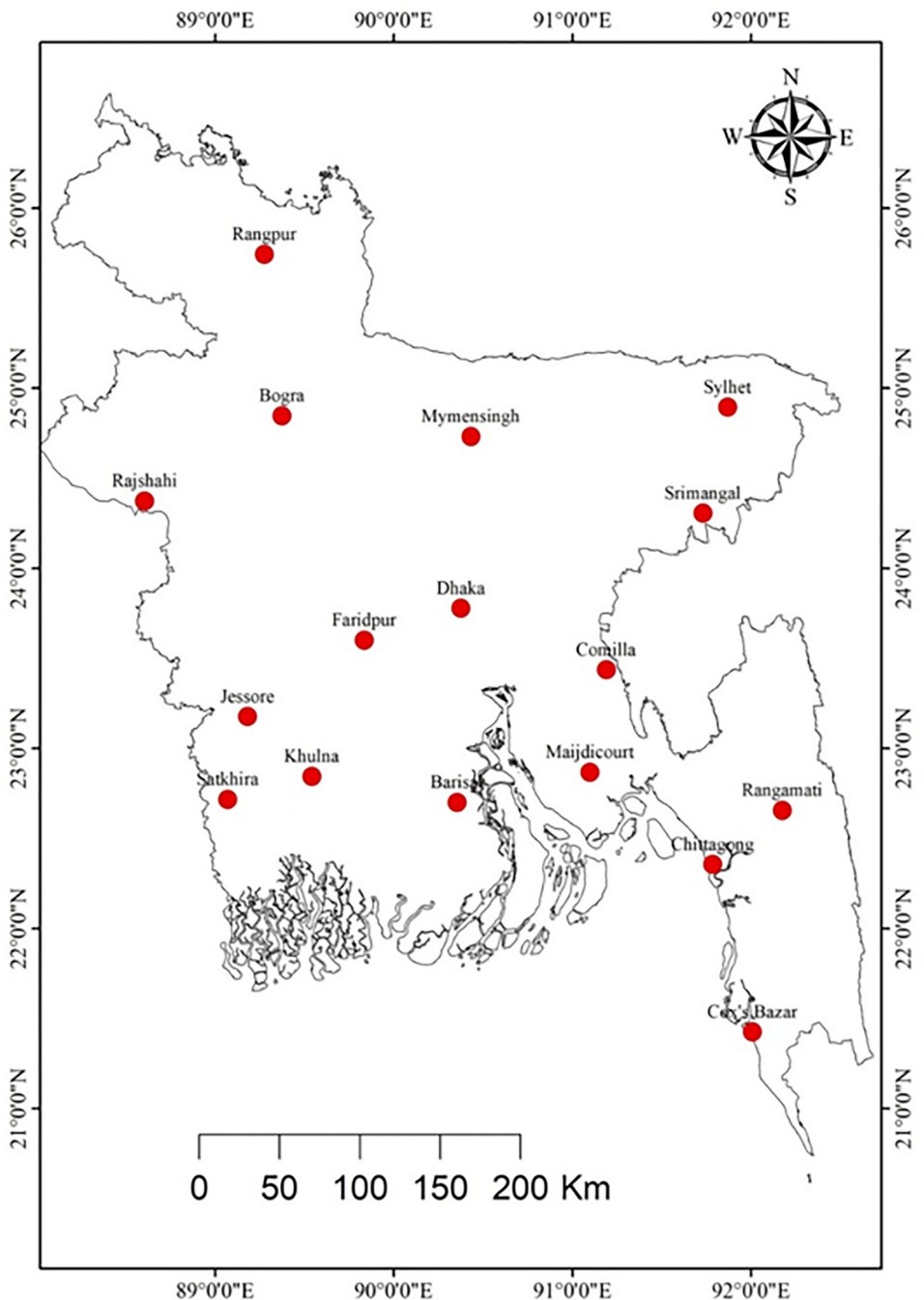

**Fig 1. Location of selected stations.**

Here, $\bar{x} = \frac{\sum_{t=1}^{n} x_t}{n}$, $n$ is the number of sample size, and $k$ is the lag. The critical value for a given level of significance (95%) is evaluated using [52] equation as,

$$r_k(95\%) = \frac{-1 \pm 1.96\sqrt{n-k-1}}{n-k}$$

**Table 1. Characteristics of stations for recorded precipitation and temperature data.**

| Station | Lat | Lon | Alt(m) | Period |
|---------|-----|-----|--------|--------|
| Cox's Bazar | 21.46 | 91.98 | 4.00 | 1949–2013 |
| Chittagong | 22.34 | 91.79 | 20.0 | 1949–2013 |
| Rangamati | 22.65 | 92.17 | 15.00 | 1957–2013 |
| Maijdicourt | 22.83 | 91.08 | 138.50 | 1949–2013 |
| Comilla | 23.48 | 91.19 | 10.00 | 1949–2013 |
| Dhaka | 23.78 | 90.39 | 9.00 | 1953–2013 |
| Faridpur | 23.61 | 89.84 | 9.00 | 1949–2013 |
| Mymensingh | 24.75 | 90.41 | 19.00 | 1949–2013 |
| Srimangal | 24.29 | 91.73 | 23.00 | 1949–2013 |
| Sylhet | 24.89 | 91.86 | 35.00 | 1956–2013 |
| Rajshahi | 24.37 | 88.60 | 18.00 | 1964–2013 |
| Rangpur | 25.72 | 89.07 | 34.00 | 1954–2013 |
| Bogra | 24.88 | 89.36 | 20.00 | 1949–2013 |
| Jessore | 23.17 | 89.22 | 7.00 | 1949–2013 |
| Khulna | 22.80 | 89.58 | 4.00 | 1949–2013 |
| Satkhira | 22.68 | 89.07 | 6.00 | 1949–2013 |
| Barisal | 22.70 | 90.36 | 10.00 | 1949–2013 |

Typically, the auto-correlation coefficient calculated at lag-1 is adequate to determine if a time series has a serial correlation if the coefficient of autocorrelation falls at a given lag between the upper and lower confidence interval limits. Otherwise, the alternative hypothesis of dependency at a 5% significance level is acknowledged. Various methods were used by other researchers to remove serial dependency like de-trending, first-order differencing, pre-whitening, variance correlation [49] and trend-free pre-whitening (TFPW) approach [53].

### Non–parametric Modified Mann–Kendall (MMK) test

The rank-based non-parametric Mann-Kendall [54,55] is used for detecting the trends in the time series data. The null hypothesis ($H_0$) in the Mann-Kendall test explains that no temperature and rainfall trends have been observed over time and the alternate hypothesis ($H_1$) explains that the trends in the series have been increasing or decreasing. Test statistics S is calculated as

$$S = \sum_{i=1}^{n-1} \sum_{j=i+1}^{n} sgn(x_j - x_i)$$

Where, $sgn(x_j-x_i)$ is the sign function given as,

$$sgn(x_j - x_i) = \begin{cases} +1, & if(x_j - x_i) > 0 \\ 0, & if(x_j - x_i) = 0 \\ -1, & if(x_j - x_i) < 0 \end{cases}$$

A very high positive $S$ value represents a sign of an increasing trend, and a very low negative value sign a decreasing trend. The variance of $S$, VAR(S), which was given by [56], is computed

as

$$Var(S) = \frac{n(n-1)(2n+5) - \sum_{i=1}^{m} t_i(t_i - 1)(2t_i + 5)}{18}$$

Where n is the number of data points, $m$ is the number of tied groups and $t_i$ is considered as the number of ties up to sample $i$. In case of sample size $n > 10$, the test statistic $Z$ is computed as

$$Z = \begin{cases} \dfrac{S-1}{\sqrt{Var(S)}}, & if \ S > 0 \\ 0, & if \ S = 0 \\ \dfrac{S+1}{\sqrt{Var(S)}}, & if \ S < 0 \end{cases}$$

Test statistics Z follows a normal distribution.

At significance level α, $Z > Z_{\propto/2}$, then the null hypothesis $H_0$ of no trend is rejected. At 5% significance level ($\alpha = 0.05$), standardised Z is equal to 1.96. If $Z$ is positive, then the trend is increasing, and a negative value indicates the trend is decreasing.

In this research, modified methods proposed by [49] have been used that incorporates the effect of serial correlation. Through multiplying it with a correction factor, the variance in MK test statistics is corrected in the proposed method. Modified variance $Var(S)^*$ is evaluated as

$$Var(S)^* = V(S)\frac{n}{n^*}$$

Where, $n^*$ is an effective sample size. The $\frac{n}{n^*}$ is given by

$$\frac{n}{n^*} = 1 + \frac{2}{n(n-1)(n-2)}$$

$$\sum_{i=1}^{n-1}(n-i)(n-i-1)(n-i-2)r_i$$

Where $n$ = actual number of observations and $r_i$ = lag $i$ is significant auto-correlation coefficient of rank $i$ of the time series. Once $Var(S)^*$ was computed then in the original MK test statistics, Var(S) is replaced by $Var(S)^*$ to calculate the Z value.

## Sen's slope

Sen's slope estimation [57] is used as non-parametric method for trend analysis which provides a robust estimation of time series trend. It is used to detect the magnitude of the trend by choosing median slope 'Q' in a sample of N pairs of data is calculated as

$$Q_i = \frac{x_j - x_k}{j - k} \ for \ i = 1, \ldots, N$$

Where $X_j$ and $X_K$ is the data values for $j$ and $k$ times of a period where $j > k$, respectively. For each observation, the slope is calculated.

The median slope ($Q'$) given by Sen's estimator of slope, as follows

$$Q' = \begin{cases} Q\lceil\dfrac{N+1}{2}\rceil & \text{if } N \text{ is odd} \\ \dfrac{Q[N+1] + Q[N+2]}{2} & \text{if } N \text{ is even} \end{cases}$$

Where $N$ is the number of the calculated slope. The two-sided test is carried out at $100(1 - \alpha)$ % of confidence interval then a true slope can be obtained by the non-parametric test in the data series. A positive value of Sen's slope indicates an upward trend, and a negative value indicates a downward trend.

## IDW interpolation

Interpolation methods are widely used in spatial analysis, and the literature contains three common interpolation techniques (Spline, IDW, and Kriging). A study [57] compared IDW, kriging, and spline spatial interpolation methods and found that IDW and kriging performed similarly and that both are more accurate than the spline interpolation method. Another study [58] indicated that the Inverse Distance Weighting (IDW) method is slightly better than the other three methods. The Inverse Distance Weighted (IDW) is a deterministic method for interpolation of multivariate sample points. The general idea of the IDW method is that it estimates unknown values by using points of known values through interpolation. The interpolating surface is a weighted average of a set of sample points. The weight is a function of the inverse distance [59]. The interpolated surface should be a location-dependent variable [60]. The nearby points will have the most influence on the interpolation, and the surface will be less smooth while the influence will decrease as distant points increase. In this study, the IDW method is used to show spatial variation in temperatures and rainfall trends.

## Regression analysis

To determine the relationship between climate variability and rice yield variability multiple linear regression analyses were performed for the period 1970 to 2013. The climatic variables and crop yields were detrended using multiple linear regression to remove the effects of non-climatic variables such as technological improvements, better crop and soil management improvements, use of fertilizer. In order to clearly represent the association between inputs or independent factors and outputs, multiple linear regression often uses parametric formulas, the parameters of which are estimated from the data. A multiple linear regression was used to estimate the association between climatic factors and crop yields [17]. Before any of the variables can be utilized in the regression analysis, they must first be normalized. To eliminate the influence of various characteristics and establish high comparability across the factors, the original continuous factors were transformed into dimensionless, homogenized numbers. The analogous equation upon normalization is known as the standard regression formula, and the coefficient is known as the standard regression coefficient, whose absolute value may represent the influence of the independent factors on the dependent factor. The main benefit of using multiple regression is that it permits us to measure the relative impact of one or more factors on the criterion values. Another benefit is that it recognizes outliers' issues. Despite its benefit, it includes a long and complicated computation and analytical process.

The coefficients derived from regression analyses indicate the sensitivity of yields in response to the growing season maximum, minimum temperature, and rainfall. The coefficient of determination ($R^2$) indicates the magnitudes of variability in rice yields can be explained by variability in climate. In addition, an augmented Dickey-Fuller (ADF) test (i.e., to

check unit-roots for each variable) [61] was performed. The input time series data must be pre-processed since hydrometeorological time series data are not stationary [35]. The non-stationary data analysis cannot provide satisfactory findings for regression analysis [62]. For that reason, the ADF test looked at unit roots in climate data [61]. Because the data series in this study spans more than 65 years, it is worth checking for stationarity [22]. Before estimating the regression model, this criterion necessitates a thorough examination of the datasets to guarantee that they are stationary. The test results revealed all the data were stationary; thus, datasets were quite prepared for further analysis of the study. The multiple linear relationship between detrended crop yield and detrended climate variables were derived as follows:

$$Y_{CY} = \alpha_0 + \alpha_1 X_{maxt} + \alpha_2 X_{mint} + \alpha_3 X_{train} + \varepsilon$$

$Y_{CY}$ represents crop yields of the dependent variables namely, Aus, Aman, Boro (in ton per hectare), $\alpha_0$ is the regression intercept, maxt is the average maximum temperature(°C) from growing season, mint is the average minimum temperature(°C) from growing season, train is the total rainfall(mm) from growing season, $\alpha_1, \alpha_2, \alpha_3$ are regression coefficients, and $\varepsilon$ is the error term.

## Results

### Changes in seasonal and annual temperatures on national level

The non-parametric Modified Mann-Kendall and Sen's slope tests were applied on aggregated monthly mean temperature ($T_{mean}$), minimum temperatures ($T_{min}$), and maximum temperature ($T_{max}$) of all the 35 stations. The results indicated an increasing trend in all months and seasons except April with significance at a 5% level of significance for $T_{mean}$. The annual $T_{mean}$ showed an increasing trend (Z value + 6.24). The annual $T_{mean}$ also increased at a rate of 0.13°C/decade. The highest $T_{mean}$ increase occurred during November at a rate of 0.24°C/decade. The post-monsoon $T_{mean}$ was higher than that of the other seasons. All of the results are captured in Table 2 below.

Table 2. Monthly mean, minimum and maximum temperatures Mann-Kendall (Z) and Sen's slope test trends results.

| Time series | Mean temperature | | Minimum temperature | | Maximum temperature | |
|---|---|---|---|---|---|---|
| | MK test | Sen's slope | MK test | Sen's slope | MK test | Sen's slope |
| January | 2.22* | 0.006* | 1.70 | 0.010 | 1.31 | 0.005 |
| February | 2.21* | 0.008* | 4.28* | 0.023* | -1.05 | -0.006 |
| March | 2.44* | 0.010* | 4.08* | 0.026* | -0.60 | 0.003 |
| April | 1.15 | 0.006 | 3.58* | 0.010* | 0.24 | 0.002 |
| May | 2.33* | 0.011* | 1.40 | 0.005 | 2.67* | 0.016* |
| June | 5.33* | 0.016* | 4.44* | 0.010* | 3.96* | 0.022* |
| July | 3.39* | 0.013* | 5.77* | 0.009* | 3.01* | 0.017* |
| August | 4.75* | 0.016* | 3.71* | 0.007* | 5.36* | 0.025* |
| September | 3.66* | 0.009* | 2.75* | 0.004* | 3.85* | 0.015* |
| October | 4.50* | 0.015* | 1.38 | 0.005 | 5.61* | 0.023* |
| November | 4.36* | 0.024* | 2.90* | 0.021* | 5.23* | 0.025* |
| December | 3.58* | 0.015* | 6.18* | 0.019* | 2.17* | 0.012* |
| Annual | 6.24* | 0.013* | 6.44* | 0.013* | 2.85* | 0.013* |
| Pre-monsoon | 2.80* | 0.010* | 3.79* | 0.014* | 1.10 | 0.006 |
| Monsoon | 3.30* | 0.014* | 5.37* | 0.007* | 5.08* | 0.022* |
| Post-monsoon | 5.65* | 0.018* | 2.77* | 0.011* | 7.05* | 0.023* |
| Winter | 3.49* | 0.010* | 4.73* | 0.018* | 0.80 | 0.002 |

* Significant trends at 5% significance level.

For the $T_{min}$, an increasing trend was observed in all months except for January, May, and October with a 5% level of significance. The annual $T_{min}$ of all stations averaged showed an increasing trend (Z value +6.44). The annual $T_{min}$ had an increase of 0.13˚C/decade. The highest $T_{min}$ increase was observed during March at a rate of 0.26˚C/decade. The $T_{min}$ in winter (December to February) was found to be significantly higher than that in the summer season (June to August). The $T_{min}$ increases in the winter season at a rate of 0.18˚C/decade. In relation to the $T_{max}$, an increasing trend from May to December with a 5% level of significance was observed while other months did not show any significant trends. The annual $T_{max}$ indicated an increasing trend (Z value +2.85). The mean annual $T_{max}$ trend is 0.13˚C /decade. The highest mean $T_{max}$ increases during November at a rate of 0.25˚C/decade.

## Changes in annual temperatures station-wise

The long-term annual $T_{mean}$ increased in all selected stations except for Rangamati and Barisal (Table 3). The highest annual $T_{mean}$ increases significantly at Sylhet station with a trend of 0.29˚C/decade. A maximum positive increase of $T_{mean}$ is also observed at Cox's Bazar and Maijdee court stations.

In the long-term, annual $T_{min}$ observed at Dhaka station indicated the highest significant increase of 0.25˚C/decade. In contrast, Rangamati station, situated in the south-eastern part of Bangladesh showed the highest significant decreasing trend of -0.28˚C/decade of annual long-term $T_{min}$ for the period (1949–2013). For long-term annual $T_{max}$ trends, Sylhet station, situated in the north-eastern part of Bangladesh showed the highest increasing trend of 0.36˚C/decade. Cox's Bazar station also recorded a significantly high positive increasing trend.

Spatial variations of temperature are depicted in Fig 2. Fig 2(A) depicts annual $T_{mean}$ with all the selected stations showed a positive significant trend except for southeastern (Rangamati), southwestern (Barisal) stations. Fig 2(B) depicts annual $T_{min}$ with all the selected

**Table 3. Annual mean, minimum and maximum temperatures Mann-Kendall (Z) and Sen's slope test trends results station wise.**

| Station | Mean temperature | | Minimum temperature | | Maximum temperature | |
|---|---|---|---|---|---|---|
| | MK | Sen's Slope | MK | Sen's slope | MK | Sen's slope |
| Cox's Bazar | 3.45* | 0.24* | 9.94* | 0.2* | 2.72* | 0.29* |
| Chittagong | 2.66* | 0.14* | 2.67* | 0.11* | 2.33* | 0.17* |
| Rangamati | -0.89 | -0.06 | -2.14* | -0.28* | 0.75 | 0.09 |
| M' court | 3.94* | 0.23* | 4.73* | 0.22* | 4.02* | 0.25* |
| Comilla | 2.18* | 0.05* | 1.48 | 0.03 | 1.67 | 0.05 |
| Dhaka | 5.15* | 0.21* | 7.2* | 0.25* | 5.14* | 0.17* |
| Faridpur | 9.2* | 0.22* | 5.35* | 0.16* | 5.88* | 0.24* |
| Srimangal | 4.12* | 0.12* | 3.98* | 0.18* | 1.63 | 0.05 |
| Sylhet | 3.65* | 0.29* | 2.45* | 0.19* | 6.13* | 0.36* |
| Mymensingh | 2.09* | 0.04* | 4.29* | 0.08* | 0.48 | 0.01 |
| Rangpur | 2.15* | 0.1* | 3.05* | 0.16* | 0.51 | 0.01 |
| Bogra | 2.62* | 0.1* | 3.86* | 0.09* | 0.92 | 0.06 |
| Rajshahi | 4.00* | 0.15* | 0.1 | 0.00 | 3.57* | 0.29* |
| Jessore | 7.67* | 0.14* | 5.65* | 0.08* | 7.14* | 0.2* |
| Khulna | 2.09* | 0.06* | -0.1 | 0.00 | 3.09* | 0.12* |
| Satkhira | 2.68* | 0.09* | 2.94* | 0.1* | 1.62 | 0.07 |
| Barisal | 1.83 | 0.08 | -0.44 | -0.02 | 5.02* | 0.16* |

* Significant trends at 5% significance level.

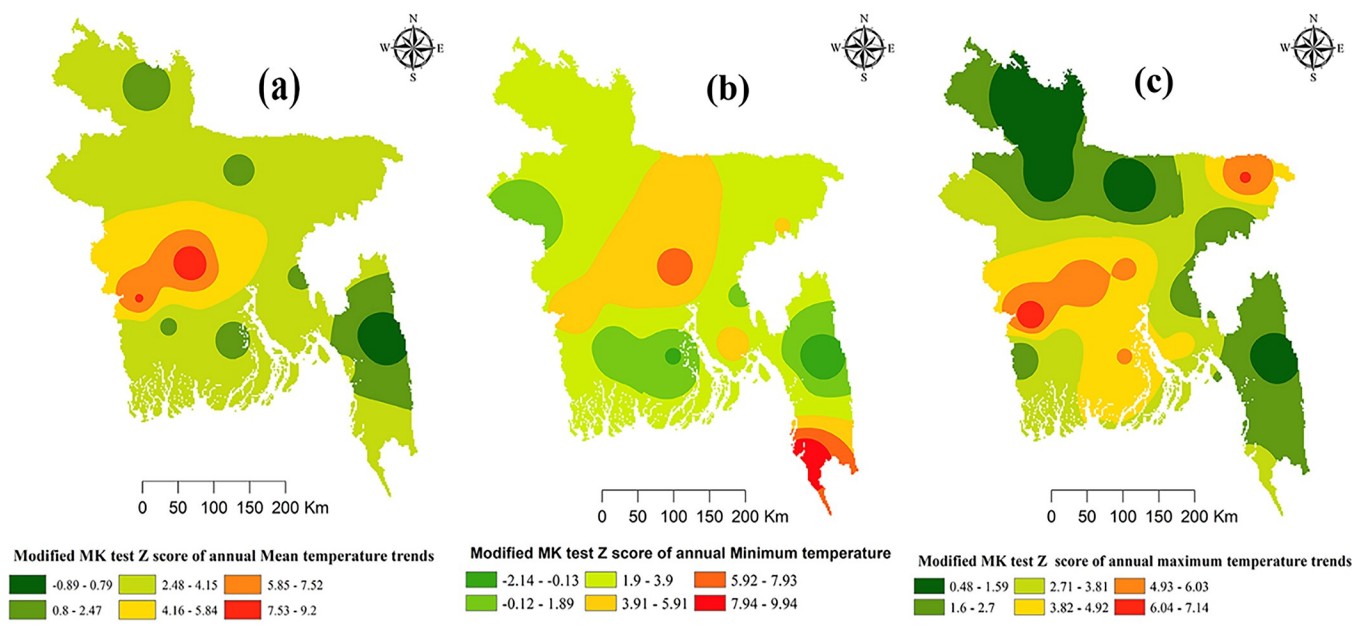

**Fig 2.** Long-term annual mean (a), mean minimum (b) and mean maximum(c) (z values from the Modified Mann-Kendall test) of Bangladesh using IDW interpolation. Fig **a** shows spatial variation of trends of annual mean temperature in all the selected stations. Fig **b** shows spatial variations of trends of annual minimum temperature in all the selected stations. Fig **c** shows spatial variations of trends of annual maximum temperature in all the selected stations.

stations showed a positive significant trend except for southeastern (Comilla), northwestern (Rajshahi), and southwestern (Khulna and Barisal). Fig 2(C) depicts annual $T_{max}$ with all the selected stations showed positive significant trend except for south-eastern (Rangamati, Comilla), north-eastern (Srimangal, Mymensingh), north-western (Rangpur, Bogra), and south-western (Satkhira) stations.

## Monthly mean, minimum and maximum temperature trends

The Modified Mann-Kendall and Sen's slope tests have been used to determine the significant trends and changes in the long-term monthly mean, minimum, and maximum temperature trends of selected stations. The results of the Mann-Kendall and Sen's slope tests on mean, minimum, and maximum temperature are presented in Tables 4–6, respectively. The test results of the monthly mean temperature showed that all the stations revealed significant increasing trends for July except for Rangamati. Data suggests that the "maximum number of stations" has increased in July and August to 16 stations, and in September and November to 15 stations (Table 4). Significantly decreasing trends were shown in January (Rangamati, Rajshahi, and Khulna), February (Khulna), March (Comilla), April (Mymensingh, Bogra), and December (Rangamati).

The test results for minimum temperature suggest a significant increasing trend for January, September, and October at five stations; April at six stations; June at 11 stations; February, March, and July at 12 stations; November and December at nine stations; and May at two stations (Table 5). In May, Cox's Bazar station depicted a significant increasing trend throughout. A significant increasing trend throughout the year except for May and September was recorded at Dhaka station. A very high increasing trend is observed in February at Dhaka station (0.54°C per decade). When the maximum number of stations was reached in January, four stations—Rangamati, Rajshahi, Khulna, and Barisal—showed a significant downward trend. Rangamati station showed a very strong decreasing trend (- 0.67°C per decade) in January.

**Table 4. Mann-Kendall (Z) and Sen's slope (SS) test results for monthly mean temperatures trends.**

| Station | Test | Jan | Feb | Mar | Apr | May | Jun | Jul | Aug | Sep | Oct | Nov | Dec |
|---|---|---|---|---|---|---|---|---|---|---|---|---|---|
| Cox's Bazar | MK | 4.01* | 5.11* | 4.09* | 3.71* | 2.96* | 4.19* | 4.61* | 4.82* | 4.12* | 4.43* | 4.80* | 4.46* |
| | SS | 0.023* | 0.032* | 0.028* | 0.022* | 0.015* | 0.022* | 0.017* | 0.025* | 0.021* | 0.027* | 0.032* | 0.030* |
| Chittagong | MK | 1.66 | 4.66* | 2.47* | 1.21 | 1.50 | 3.29* | 3.16* | 4.19* | 2.86* | 2.84* | 3.24* | 2.82* |
| | SS | 0.008 | 0.018* | 0.013* | 0.009 | 0.008 | 0.014* | 0.014* | 0.017* | 0.011* | 0.016* | 0.024* | 0.023* |
| Rangamati | MK | -2.26* | -1.49 | -1.92 | 0.00 | -0.86 | 0.53 | 1.49 | 1.37 | -1.35 | -1.17 | -1.52 | -1.96* |
| | SS | -0.029* | -0.023 | -0.013 | 0.000 | -0.007 | 0.004 | 0.007 | 0.007 | -0.006 | 0.005 | -0.016 | -0.018* |
| M' court | Mk | 2.06* | 2.02* | 2.99* | 1.89 | 2.44* | 5.39* | 4.54* | 3.22* | 4.80* | 5.39* | 4.63* | 3.71* |
| | SS | 0.012* | 0.017* | 0.018* | 0.014 | 0.018* | 0.033* | 0.030* | 0.030* | 0.022* | 0.031* | 0.040* | 0.027* |
| Comilla | MK | -0.31 | -0.37 | -2.18* | -1.68 | 0.18 | 3.70* | 4.46* | 4.26* | 3.11* | 3.33* | 2.15* | 1.76 |
| | SS | -0.001 | -0.002 | -0.016* | -0.011 | 0.000 | 0.018* | 0.019* | 0.020* | 0.014* | 0.014* | 0.014* | 0.011 |
| Dhaka | MK | 3.38* | 2.60* | 2.24* | 0.59 | 1.00 | 3.45* | 4.74* | 5.88* | 3.98* | 5.76* | 4.91* | 8.19* |
| | SS | 0.022* | 0.022* | 0.018* | 0.004 | 0.007 | 0.026* | 0.023* | 0.026* | 0.017* | 0.031* | 0.040* | 0.035* |
| Faridpur | MK | 2.46* | 2.04* | 2.92* | 3.25* | 3.15* | 6.95* | 4.96* | 5.76* | 4.33* | 4.51* | 4.70* | 6.32* |
| | SS | 0.016* | 0.017* | 0.023* | 0.023* | 0.022* | 0.020* | 0.022* | 0.025* | 0.016* | 0.021* | 0.032* | 0.032* |
| Mymensingh | MK | 0.97 | 0.63 | -0.74 | -2.49* | 0.47 | 1.93 | 2.24* | 3.23* | 2.99* | 2.14* | 2.33* | 0.81 |
| | SS | 0.006 | 0.004 | 0.005 | -0.017* | 0.003 | 0.010 | 0.011* | 0.015* | 0.014* | 0.011* | 0.014* | 0.004 |
| Srimangal | MK | 0.81 | 1.52 | 1.73 | -0.40 | -0.10 | 1.58 | 3.50* | 3.48* | 3.47* | 2.60* | 2.92* | 4.40* |
| | SS | 0.004 | 0.013 | 0.012 | -0.002 | 0.000 | 0.006 | 0.013* | 0.014* | 0.014* | 0.015* | 0.020* | 0.029* |
| Sylhet | MK | 4.55* | 4.46* | 3.34* | 1.16 | 3.35* | 2.80* | 3.75* | 4.02* | 3.15* | 4.89* | 7.09* | 4.91* |
| | SS | 0.035* | 0.040* | 0.029* | 0.010 | 0.023* | 0.021* | 0.026* | 0.027* | 0.026* | 0.035* | 0.041* | 0.035* |
| Rajshahi | MK | -2.80* | 0.40 | 2.42* | 3.76* | 4.03* | 2.22* | 3.69* | 5.30* | 2.96* | 1.44 | -0.32 | 0.33 |
| | SS | -0.027* | 0.005 | 0.025* | 0.040* | 0.044* | 0.023* | 0.024* | 0.028* | 0.018* | 0.010 | -0.003 | 0.003 |
| Rangpur | MK | 1.41 | 1.90 | 1.32 | -0.21 | 1.63 | 1.75 | 2.85* | 3.07* | 1.85 | 3.35* | 2.76* | 3.46* |
| | SS | 0.008 | 0.017 | 0.005 | -0.001 | 0.015 | 0.011 | 0.014* | 0.016* | 0.012 | 0.017* | 0.018* | 0.018* |
| Bogra | MK | 0.55 | -0.05 | 0.07 | -2.07* | 1.96* | 3.28* | 4.38* | 2.33* | 5.59* | 4.79* | 3.33* | 2.94* |
| | SS | 0.002 | -0.000 | 0.000 | -0.023* | 0.022* | 0.020* | 0.026* | 0.031* | 0.027* | 0.028* | 0.020* | 0.016* |
| Jessore | MK | 0.07 | 0.47 | 2.17* | 5.33* | 3.34* | 4.70* | 5.06* | 5.47* | 2.70* | 1.90 | 3.30* | 1.07 |
| | SS | 0.000 | 0.004 | 0.015* | 0.027* | 0.022* | 0.022* | 0.019* | 0.019* | 0.019* | 0.010 | 0.021* | 0.006 |
| Khulna | MK | -2.30* | -2.12* | 0.30 | 1.68 | 1.89 | 3.36* | 3.64* | 3.58* | 2.79* | 1.91 | 2.37* | -1.37 |
| | SS | -0.016* | -0.015* | 0.002 | 0.010 | 0.011 | 0.017* | 0.018* | 0.016* | 0.011* | 0.010 | 0.013* | -0.007 |
| Satkhira | MK | -0.11 | -0.20 | 1.01 | 2.08* | 2.21* | 2.82* | 3.83* | 3.10* | 4.50* | 1.25 | 3.13* | -0.06 |
| | SS | 0.000 | -0.001 | 0.006 | 0.115* | 0.013* | 0.013* | 0.016* | 0.017* | 0.010* | 0.006 | 0.020* | 0.000 |
| Barisal | MK | -0.07 | 0.17 | 1.02 | -0.01 | 1.55 | 5.42* | 2.62* | 2.97* | 2.31* | 2.93* | 3.29* | 1.45 |
| | SS | 0.000 | 0.000 | 0.007 | 0.000 | 0.006 | 0.022* | 0.011* | 0.015* | 0.008* | 0.012* | 0.019* | 0.010 |

* Significant trends at 5% significance level.

The test results for maximum temperature trends show that in August, all the selected stations showed significant increasing trends. The findings suggest that the maximum numbers of significant increasing trends were shown for June in 12 stations, July in 16 stations, August in all stations, September in 14 stations, October in 15 stations, November in 14 stations, and December in 10 stations (Table 6). As such, it is evidenced that all the selected stations are experiencing a significant increasing trend from June to December. In April, the five stations of Comilla, Mymensingh, Srimangal, Rangpur, and Bogra exhibited a significant decreasing trend. Cox's Bazar observed a significant increasing trend throughout the year, and a significant increase occurred between August and December. In the Sylhet station, a significant increasing trend was recorded throughout the year except for March and April, and a

**Table 5. Mann-Kendall (Z) and Sen's slope (SS) test results for monthly minimum temperatures trends.**

| Station | Test | Jan | Feb | Mar | Apr | May | Jun | Jul | Aug | Sep | Oct | Nov | Dec |
|---|---|---|---|---|---|---|---|---|---|---|---|---|---|
| Cox's Bazar | MK | 5.19* | 5.86* | 4.91* | 3.74* | 1.49 | 4.41* | 3.98* | 4.68* | 5.30* | 4.37* | 4.47* | 4.82* |
| | SS | 0.024* | 0.036* | 0.033* | 0.023* | 0.007 | 0.014* | 0.015* | 0.012* | 0.011* | 0.015* | 0.027* | 0.025* |
| Chittagong | MK | 1.27 | 4.35* | 2.06* | 2.10* | -0.50 | 2.21* | 3.72* | 2.90* | 2.68* | 1.54 | 2.68* | 4.00* |
| | SS | 0.006 | 0.018* | 0.017* | 0.015* | 0.1 | 0.007* | 0.011* | 0.011* | 0.008* | 0.006 | 0.020* | 0.017* |
| Rangamati | MK | -2.25* | -3.17* | -2.25* | -1.46 | -2.32* | -1.29 | -0.91 | -0.53 | -2.64* | -3.84* | -1.79 | -1.86 |
| | SS | -0.067* | -0.053* | -0.028* | -0.017 | -0.019* | -0.005 | -0.004 | -0.006 | -0.017* | -0.020* | -0.042 | -0.048 |
| M' court | Mk | 5.12* | 3.88* | 2.31* | 0.93 | -0.16 | 3.50* | 4.11* | 2.97* | 4.30* | 3.70* | 5.03* | 8.28* |
| | SS | 0.027* | 0.026* | 0.030* | 0.006 | 0.000 | 0.016* | 0.018* | 0.017* | 0.013* | 0.017* | 0.046* | 0.044* |
| Comilla | MK | -1.24 | 2.68* | 1.46 | -1.31 | -1.98* | 1.85 | 3.88* | 4.42* | 1.28 | 0.10 | 0.83 | 0.19 |
| | SS | -0.008 | 0.018* | 0.011 | -0.007 | -0.010* | 0.008 | 0.010* | 0.010* | 0.003 | 0.000 | 0.008 | 0.000 |
| Dhaka | MK | 8.78* | 7.53* | 3.74* | 2.57* | 0.46 | 3.43* | 3.60* | 3.59* | 1.65 | 3.01* | 4.96* | 18.89* |
| | SS | 0.042* | 0.054* | 0.045* | 0.013* | 0.003 | 0.015* | 0.010* | 0.010* | 0.006 | 0.017* | 0.043* | 0.052 |
| Faridpur | MK | 1.27 | 4.38* | 11.93* | 2.69* | 2.03* | 4.03* | 3.58* | 1.98* | 0.98 | 1.35 | 3.19* | 3.44* |
| | SS | 0.010 | 0.033* | 0.035* | 0.016* | 0.011* | 0.016* | 0.015* | 0.006* | 0.003 | 0.007 | 0.025* | 0.022* |
| Mymensingh | MK | 0.85 | 4.06* | 3.44* | 0.68 | 1.75 | 3.23* | 4.51* | 4.88* | 2.22* | -1.14 | -1.18 | -1.46 |
| | SS | 0.005 | 0.029* | 0,025* | 0.003 | 0.007 | 0.014* | 0.011* | 0.012* | 0.007* | 0.005 | 0.007 | 0.006 |
| Srimangal | MK | 3.63* | 3.99* | 3.02* | 6.08* | 1.36 | 3.25* | 5.23* | 4.60* | 4.48* | 1.94 | 3.41* | 5.62* |
| | SS | 0.031* | 0.032* | 0.032* | 0.010* | 0.001 | 0.010* | 0.010* | 0.010* | 0.011* | 0.012 | 0.027* | 0.039* |
| Sylhet | MK | 2.33* | 4.09* | 3.78* | 1.05 | 1.76 | 1.83 | 1.80 | 2.66* | 1.58 | 3.45* | 4.19* | 4.27* |
| | SS | 0.019* | 0.032* | 0.035* | 0.008 | 0.011 | 0.006 | 0.013 | 0.013* | 0.013 | 0.021* | 0.034* | 0.031* |
| Rajshahi | MK | -3.19* | 0.37 | 1.17 | 0.47 | 0.26 | 0.86 | 1.88 | 2.42* | 0.27 | -1.54 | -1.18 | -0.33 |
| | SS | -0.032* | 0.002 | 0.012 | 0.003 | 0.000 | 0.004 | 0.010 | 0.010* | 0.000 | −0.007 | -0.016 | -0.000 |
| Rangpur | MK | 1.89 | 2.97* | 3.78* | 2.12* | 2.77* | 2.20* | 2.59* | 0.62 | 0.56 | 3.09* | 2.82* | 3.88* |
| | SS | 0.013 | 0.030* | 0.031* | 0.015* | 0.013* | 0,010* | 0.006* | 0.000 | 0.0007 | 0.014* | 0.019* | 0.025* |
| Bogra | MK | -0.38 | 3.22* | 4.83* | 0.34 | 0.67 | 2.01* | 1.91 | 3.73* | 1.39 | 1.38 | 1.48 | 1.70 |
| | SS | -0.000 | 0.022* | 0.033* | 0.00 | 0.002 | 0.006* | 0.006 | 0.006* | 0.005 | 0.004 | 0.012 | 0.010 |
| Jessore | MK | -0.33 | 2.51* | 2.12* | 0.88 | -0.01 | 4.99* | 4.20* | 3.05* | 1.19 | -0.52 | 1.69 | 2.18 |
| | SS | -0.002 | 0.020* | 0.187* | 0.005 | 0.0 | 0.011* | 0.009* | 0.006* | 0.000 | -0.003 | 0.014 | 0.013 |
| Khulna | MK | -2.91* | -1.51 | 0.08 | 0.19 | -0.70 | 1.09 | 3.12* | 2.30* | 0.90 | -0.66 | 0.82 | -0.83 |
| | SS | -0.027* | -0.010 | 0.000 | 0.000 | -0.002 | 0.004 | 0.008* | 0.004* | 0.00 | -0.002 | 0.006 | -0.008 |
| Satkhira | MK | -0.30 | 1.66 | 2.97* | 1.17 | 0.32 | 4.31* | 5.78* | 5.13* | 1.24 | -0.44 | 2.87* | 1.34 |
| | SS | -0.002 | 0.011 | 0.014* | 0.005 | 0.000 | 0.017* | 0.013* | 0.011* | 0.005 | -0.000 | 0.020* | 0.006 |
| Barisal | MK | -2.87* | -1.09 | 0.52 | -1.09 | -1.15 | 0.47 | 0.47 | 0.43 | -0.74 | -1.42 | 0.73 | -0.62 |
| | SS | -0.020* | -0.008 | 0.005 | -0.008 | -0.006 | 0.002 | 0.000 | 0.000 | -0.004 | -0.007 | 0.003 | -0.006 |

* Significant trends at 5% significance level.

significant increase occurred from July to January. The stations of Comilla and Bogra showed a consecutively significant decreasing trend in February, March, and April. The Rajshahi station has the highest increasing trend, with May recording 0.84°C per decade and April recording 0.70°C per decade. The highest decreasing trend is observed in Bogra in March, recording -0.42°C per decade.

## Changes in annual and seasonal rainfall on national level

The Modified Mann-Kendall and Sen's slope tests were applied on the mean annual rainfall for all the 35 stations to obtain monthly and seasonal trends. The result shown in Table 7 indicated a significant increasing trend in the month of May, while other months did not show any

**Table 6. Mann-Kendall (Z) and Sen's slope (SS) test results for monthly maximum temperatures trends.**

| Station | Test | Jan | Feb | Mar | Apr | May | Jun | Jul | Aug | Sep | Oct | Nov | Dec |
|---|---|---|---|---|---|---|---|---|---|---|---|---|---|
| Cox's Bazar | MK | 2.30* | 3.36* | 2.47* | 2.61* | 3.57* | 4.59* | 3.20* | 4.17* | 3.56* | 3.59* | 5.19* | 3.46* |
| | SS | 0.020* | 0.028* | 0.024* | 0.023* | 0.023* | 0.031* | 0.025* | 0.040* | 0.030* | 0.038* | 0.037* | 0.033* |
| Chittagong | MK | 1.96* | 2.13* | 1.66 | 0.65 | 1.66 | 4.81* | 3.20* | 4.51* | 2.15* | 3.12* | 4.11* | 2.33* |
| | SS | 0.015* | 0.019* | 0.013 | 0.003 | 0.014 | 0.021* | 0.025* | 0.024* | 0.017* | 0.027* | 0.029* | 0.027* |
| Rangamati | MK | -0.07 | -0.09 | -1.07 | 0.78 | -0.32 | 0.94 | 2.10* | 2.14* | 0.42 | 1.26 | 2.16* | 0.33 |
| | SS | 0.000 | 0.000 | -0.01 | 0.015 | 0.000 | 0.008 | 0.020* | 0.02* | 0.002 | 0.009 | 0.011* | 0.006 |
| M' Court | Mk | 0.64 | -0.05 | 1.33 | 1.95 | 3.14* | 5.08* | 4.61* | 3.80* | 4.09* | 7.80* | 3.82* | 1.31 |
| | SS | 0.005 | 0.00 | 0.009 | 0.020 | 0.035* | 0.053* | 0.043* | 0.043* | 0.030* | 0.043* | 0.036* | 0.010 |
| Comilla | MK | 0.18 | -2.50* | -3.77* | -2.12* | 1.08 | 2.97* | 4.22* | 3.53* | 2.92* | 3.79* | 3.56* | 2.87* |
| | SS | 0.000 | -0.027* | -0.039* | -0.016* | 0.009 | 0.026* | 0.025* | 0.029* | 0.023* | 0.029* | 0.02* | 0.020* |
| Dhaka | MK | 0.29 | -1.09 | -1.05 | -0.62 | 1.38 | 6.36* | 4.69* | 5.42* | 5.17* | 5.01* | 3.63* | 2.33* |
| | SS | 0.0 | -0.01 | -0.008 | -0.002 | 0.010 | 0.035* | 0.038* | 0.042* | 0.030* | 0.044* | 0.041* | 0.018* |
| Faridpur | MK | 2.57* | -0.04 | 0.70 | 2.79* | 3.07* | 3.45* | 4.19* | 5.57* | 4.31* | 8.28* | 4.92* | 4.89* |
| | SS | 0.022* | 0.00 | 0.005 | 0.027* | 0.02* | 0.021* | 0.029* | 0.044* | 0.027* | 0.036* | 0.038* | 4.00* |
| Mymensingh | MK | 0.13 | -1.66 | -2.98* | -4.64* | -0.48 | 0.83 | 1.55 | 2.33* | 5.61* | 3.68* | 4.54* | 2.39* |
| | SS | 0.00 | -0.018 | -0.032* | -0.037* | 0.002 | 0.004 | 0.007 | 0.017* | 0.020* | 0.029* | 0.035* | 0.014* |
| Srimangal | MK | 0.86 | -0.34 | -0.86 | -2.00* | -0.84 | 0.72 | 2.55* | 3.64* | 2.08* | 2.47* | 2.19* | 1.98* |
| | SS | 0.006 | -0.005 | -0.008 | -0.013* | -0.004 | 0.003 | 0.016* | 0.018* | 0.016* | 0.017* | 0.018 | 0.016* |
| Sylhet | MK | 4.64* | 3.56* | 1.55 | 0.58 | 3.39* | 2.97* | 3.94* | 4.51* | 4.43* | 4.52* | 5.44* | 5.17* |
| | SS | 0.046* | 0.05* | 0.024 | 0.006 | 0.031* | 0.035* | 0.040* | 0.040* | 0.038* | 0.040* | 0.050* | 0.041* |
| Rajshahi | MK | -1.84 | 0.07 | 2.28* | 2.92* | 5.55* | 2.11* | 3.35* | 4.34* | 3.36* | 6.36* | 1.07 | 1.95 |
| | SS | -0.019 | 0.00 | 0.037* | 0.070* | 0.084* | 0.042* | 0.037* | 0.046* | 0.037* | 0.035* | 0.009 | 0.011 |
| Rangpur | MK | 0.45 | 0.29 | -2.16* | -2.20* | 0.99 | 0.62 | 2.27* | 3.03* | 1.73 | 2.57* | 1.85 | 1.73 |
| | SS | 0.000 | 0.00 | -0.02* | -0.023* | 0.012 | 0.008 | 0.018* | 0.027* | 0.016 | 0.017* | 1.73 | 0.010 |
| Bogra | MK | 1.23 | -2.64* | -2.24* | -2.39* | 1.71 | 2.70* | 4.63* | 5.16* | 4.77* | 5.86* | 2.86* | 1.98* |
| | SS | 0.009 | -0.023* | -0.042* | -0.04* | 0.036 | 0.038* | 0.044* | 0.055* | 0.046* | 0.048* | 0.031* | 0.022* |
| Jessore | MK | -0.01 | -1.66 | 7.14* | 3.53* | 5.01* | 2.79* | 4.15* | 4.00* | 2.83* | 3.50* | 5.17* | 0.11 |
| | SS | 0.000 | -0.018 | 0.015* | 0.037* | 0.036* | 0.033* | 0.027* | 0.033* | 0.019* | 0.023* | 0.029* | 0.000 |
| Khulna | MK | -1.15 | -1.51 | 0.20 | 1.98* | 3.38* | 4.33* | 3.71* | 4.62* | 4.40* | 3.25* | 2.10* | -0.85 |
| | SS | -0.010 | -0.019 | 0.000 | 0.017* | 0.019* | 0.030* | 0.025* | 0.027* | 0.021* | 0.023* | 0.019* | -0.004 |
| Satkhira | MK | -0.25 | -2.04* | -0.72 | 1.73 | 2.63* | 1.59 | 2.83* | 2.82* | 2.23* | 1.89 | 1.38 | -1.20 |
| | SS | 0.000 | -0.019* | -0.005 | 0.012 | 0.021* | 0.012 | 0.017* | 0.023* | 0.013* | 0.011 | 0.017 | -0.010 |
| Barisal | MK | 2.22* | 1.07 | 1.22 | 0.76 | 4.22* | 4.28* | 2.51* | 4.50* | 3.53* | 6.90* | 5.00* | 3.36* |
| | SS | 0.017* | 0.006 | 0.010 | 0.004 | 0.013* | 0.037* | 0.017* | 0.025* | 0.020* | 0.029* | 0.033* | 0.021* |

* Significant trends at 5% significance level.

significant trends. The annual mean rainfall of all stations showed an increasing trend (Z value +2.03). Analyzing the trend by Sen's estimator reveals that the annual mean rainfall increased at a rate of 4.20 mm/year, pre-monsoon rainfall increased significantly (Z value +2.39) at a rate of 1.35 mm/year while monsoon, post-monsoon and winter rainfall did not show any significant trends. Long-term seasonal and annual variation in rainfall is presented in Fig 3.

## Changes in annual and seasonal rainfall trends station-wise

Annual rainfall trends identified, a significant increasing trend of 5.82 mm/year in Khulna station, while the highest significant decreasing trend -8.92 mm/year was observed in Comilla

Table 7. Monthly and seasonal rainfall test results at national level.

| Time series | MK test | Sen's slope |
| --- | --- | --- |
| January | -0.52 | -0.008 |
| February | 0.70 | 0.034 |
| March | -0.63 | -0.069 |
| April | 0.76 | 0.30 |
| May | 2.06* | 1.05* |
| June | -0.31 | -0.29 |
| July | 1.48 | 0.92 |
| August | 0.49 | 0.24 |
| September | 1.88 | 0.81 |
| October | 0.85 | 0.56 |
| November | -0.82 | -0.09 |
| December | 0.10 | 0.00 |
| Annual | 2.03* | 4.20* |
| Pre-monsoon | 2.39* | 1.35* |
| Monsoon | 1.72 | 2.23 |
| Post-monsoon | 0.74 | 0.50 |
| winter | 0.92 | 0.13 |

* Significant trends at 5% significance level.

station during 1949–2013 (Table 8). Rajshahi station also showed a significant decreasing trend, but no significant annual rainfall trends were observed at other stations. On the other hand, pre-monsoon rainfall is increasing at six stations (Cox's Bazar, Chittagong, Rangamati, Mymensingh, Rangpur, Satkhira) while other stations did not show any significant trends. For Pre-monsoon rainfall, Rangamati and Cox's Bazar stations showed the highest significant increasing trend of 4.11mm/year. No significant increasing monsoon rainfall was observed at any station, while four stations (Maijdee Court, Comilla, Srimangal, and Rajshahi) showed significant decreasing trends. The highest significant decreasing trend is observed at Comilla station, where monsoon rainfall is decreasing at a rate of -8.90 mm/year. No selected stations showed a significant increase or decrease for post-monsoon rainfall of the selected stations. Only the south-western Khulna station showed a significant increasing trend for winter rainfall.

## Monthly trends for rainfall

The monthly trends of the data from the selected 17 stations depicted a varied pattern in Bangladesh (Table 9). Significant increasing trends were showed in February (Khulna), April (Cox's Bazar), May (Cox's Bazar, Chittagong, Rangamati), July (Rangpur), September (Jessore, Khulna) and Rangpur station showed significant increasing trends for the month October, November, and December, consecutively. Significant decreasing trends were shown in January (Sylhet), March (Bogra), June (Comilla), July (Comilla, Rajshahi), August (Comilla), other months did not show any significant decreasing trend. The highest monthly significant increasing trend was shown at Cox's Bazar station in May at a rate of 2.75 mm/month. The highest monthly significant decreasing trend showed at Comilla and Rajshahi station in July at a rate of—3.34 mm/month. The north-western Rangpur station showed maximum significant increasing trends for the months of July, August, October, Novermber, and December. While, south-eastern Comilla station has maximum significant decreasing trends that occurred in the three consecutive months of June, July, and August.

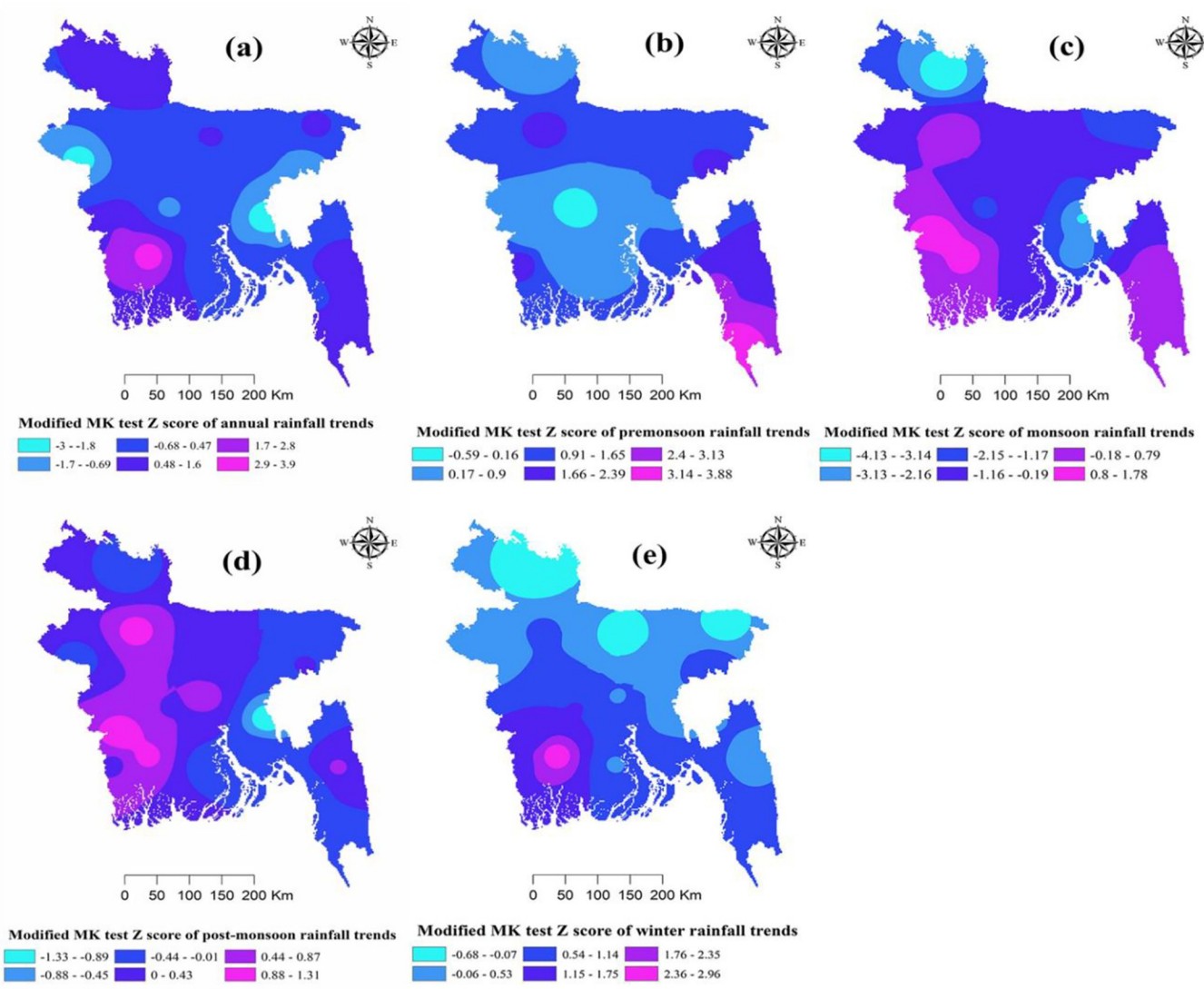

**Fig 3. Long-term annual and seasonal rainfall (Z values from the Modified Mann-Kendall test) of Bangladesh using IDW interpolation.** Fig **3a** shows spatial variations of trends of annual rainfall, Fig **3b** shows spatial variations of trends of pre-monsoon rainfall, Fig **3c** shows spatial variations of trends of monsoon rainfall, Fig **3d** shows spatial variations of trends of post-monsoon rainfall, and **3e** shows spatial variations of trends of winter rainfall trends.

## Climate and rice yield relationship

To determine the relationship between climatic variability and rice yields, a pearson correlation analysis was performed. The results suggest that maximum temperature and minimum temperature had a significant correlation in the yield of three rice crops. The highest strong and positive correlation is found between maximum temperature and *Aman* yield (+0.75). A significant moderate correlation is found between minimum temperature and *Aus* yield (+0.56). In terms of rainfall, a significant moderate and positive correlation (+0.62) was found between rainfall and Aman yield, whereas a significant positive weak correlation between rainfall and Aus yield (+0.37). There was no significant effect of rainfall on the yield of Boro yield (+0.15). The correlation between the climatic variables and the crop yields are provided in Table 10.

**Table 8. Long-term annual and seasonal rainfall trends results station-wise.**

| Station | Test | Pre-monsoon | Monsoon | Post-monsoon | Winter | Annual |
|---|---|---|---|---|---|---|
| Cox's Bazar | MK | 3.88* | 0.47 | -0.19 | 0.64 | 1.12 |
| | SS | 4.11* | 1.50 | -0.27 | 0.05 | 3.45 |
| Chittagong | MK | 2.71* | 0.01 | -0.26 | 0.62 | 0.18 |
| | SS | 2.47* | 0.05 | -0.29 | 0.06 | 0.88 |
| Rangamati | MK | 2.01* | 0.43 | 0.50 | 0.46 | 1.43 |
| | SS | 4.41* | 0.86 | 0.47 | 0.00 | 6.33 |
| M'Court | Mk | 1.35 | -2.85* | -0.03 | 0.71 | -0.52 |
| | SS | 1.90 | -3.54* | -0.05 | 0.06 | -1.65 |
| Comilla | MK | 0.73 | -3.25* | -1.33 | 0.28 | -3.00* |
| | SS | 0.91 | -8.90* | -0.89 | 0.00 | -8.92* |
| Dhaka | MK | 0.48 | -0.44 | 0.60 | 0.50 | 0.05 |
| | SS | 0.89 | -1.44 | 0.42 | 0.08 | 0.42 |
| Faridpur | MK | -0.59 | -1.57 | 0.41 | 1.16 | -1.04 |
| | SS | -0.76 | -2.84 | 0.26 | 0.15 | -2.61 |
| Mymensingh | MK | 1.97* | -0.21 | 0.02 | 0.92 | 0.57 |
| | SS | 2.29* | -0.42 | 0.04 | 0.11 | 3.00 |
| Srimangal | MK | 1.09 | -1.98* | -0.36 | -0.38 | -1.2 |
| | SS | 2.21 | -3.36* | -0.36 | -0.06 | -2.47 |
| Sylhet | MK | 1.56 | -0.35 | 0.17 | -0.48 | 0.47 |
| | SS | 3.72 | -1.64 | 0.15 | -0.10 | 3.28 |
| Rajshahi | MK | 0.53 | -4.13* | -0.31 | -0.68 | -2.36* |
| | SS | 0.37 | -7.33* | -0.25 | -0.15 | -7.23* |
| Rangpur | MK | 2.13* | 0.64 | 1.12 | 0.64 | 1.54 |
| | SS | 2.27* | 0.05 | 1.13 | 0.05 | 4.64 |
| Bogra | MK | 0.92 | -0.20 | -0.23 | -0.00 | 0.39 |
| | SS | 0.96 | -0.57 | -0.15 | 0.00 | 1.09 |
| Jessore | MK | 0.33 | 1.78 | 1.31 | 1.24 | 1.69 |
| | SS | 0.24 | 3.32 | 0.58 | 0.19 | 4.12 |
| Khulna | MK | 0.45 | 1.41 | 1.05 | 2.96* | 3.94* |
| | SS | 0.36 | 2.86 | 0.67 | 0.33* | 5.82* |
| Satkhira | MK | 2.30* | 0.42 | 0.31 | 1.63 | 1.65 |
| | SS | 1.21* | 0.93 | 0.13 | 0.35 | 3.00 |
| Barisal | MK | 0.41 | -0.58 | -0.41 | 0.45 | -0.46 |
| | SS | 0.22 | -1.18 | -0.44 | 0.06 | -1.33 |

* Significant trends at 5% significance level. Sen's slope unit is in mm.

## Climate explained yield variability

To determine the relationship between climate variability and rice yield variability a multiple linear regression analyse was performed. The coefficient of determination ($R^2$) indicates the magnitudes of variability in rice yields can be explained by variability in climate. The p-value of F-statistics indicates the statistical significance of three models. The test results are shown in Table 11.

The results suggest that 33% of Aus yield variability could be explained by growing season climate variability, whereas the remaining 67% of yield variability influenced by non-climatic factors. Similarly, the coefficient of determination ($R^2$) from *Aman* suggests that 25% of *Aman* yield variability can be explained by climate variability. In addition, only 16% of the variation

**Table 9. Mann-Kendall (Z) and Sen's slope (SS) test results for monthly rainfall trends.**

| Station | Test | Jan | Feb | Mar | Apr | May | Jun | July | Aug | Sep | Oct | Nov | Dec |
|---|---|---|---|---|---|---|---|---|---|---|---|---|---|
| Cox's Bazar | MK | 0.34 | 0.18 | 0.53 | 2.05* | 2.29* | 0.87 | -0.62 | 0.48 | 0.45 | -0.52 | 0.16 | 0.22 |
| | SS | 0.00 | 0.00 | 0.00 | 0.93* | 2.75* | 1.74 | -1.45 | 1.00 | 0.59 | -0.45 | 0.00 | 0.00 |
| Chittagong | MK | -1.23 | 0.20 | -0.42 | 1.46 | 2.44* | -0.05 | -0.00 | -0.84 | 0.05 | 0.00 | 0.41 | 0.56 |
| | SS | 0.00 | 0.00 | 0.00 | 0.76 | 1.93* | -0.06 | 0.00 | -1.40 | 0.06 | 0.00 | 0.00 | 0.00 |
| Rangamati | MK | 0.15 | 1.16 | 1.09 | 1.53 | 2.62* | 0.46 | -0.21 | -0.28 | 0.89 | 0.20 | 1.05 | -0.05 |
| | SS | 0.00 | 0.0 | 0.06 | 0.96 | 2.52* | 0.74 | -0.39 | 1.16 | 0.73 | 0.22 | 0.19 | 0.00 |
| M' court | Mk | -1.05 | 0.76 | 0.67 | 0.48 | 0.90 | -1.21 | -0.61 | -0.92 | -0.26 | 0.02 | -0.56 | 1.07 |
| | SS | 0.00 | 0.00 | 0.08 | 0.32 | 0.82 | -1.61 | -1.05 | -1.23 | -0.2 | 0.02 | 0.00 | 0.00 |
| Comilla | MK | -0.40 | 0.37 | -0.27 | -0.48 | 1.23 | -2.34* | -2.40* | -2.90* | -1.55 | 0.20 | 1.05 | -0.05 |
| | SS | 0.00 | 0.00 | -0.05 | -0.28 | 1.11 | -2.85* | -3.34* | -2.77* | -1.41 | 0.22 | 0.19 | 0.00 |
| Dhaka | MK | -0.43 | 0.14 | 0.56 | 0.47 | 0.21 | -1.02 | -0.20 | 0.00 | 0.41 | 0.72 | -0.62 | 0.50 |
| | SS | 0.00 | 0.00 | 0.12 | 0.28 | 0.33 | 0.99 | -0.22 | 0.00 | 0.37 | 0.59 | 0.00 | 0.00 |
| Faridpur | MK | 0.70 | 0.87 | -0.56 | -0.57 | -0.46 | -1.30 | -1.41 | -0.45 | -0.58 | 0.32 | -0.58 | 1.16 |
| | SS | 0.00 | 0.00 | 0.05 | 0.32 | -0.25 | -1.13 | -1.17 | -0.34 | -0.42 | 0.23 | 0.00 | 0.00 |
| Mymensingh | MK | -1.04 | 0.83 | 0.005 | 2.88* | 1.20 | -1.13 | 1.49 | 0.03 | 0.18 | 0.05 | -1.59 | 0.73 |
| | SS | 0.00 | 0.00 | 0.00 | 1.30* | 1.05 | -1.32 | 1.53 | 0.02 | 0.19 | 0.03 | 0.00 | 0.00 |
| Srimangal | MK | -1.43 | 0.00 | 0.00 | -0.50 | 1.80 | -1.73 | -0.54 | 0.04 | 0.00 | 0.03 | -1.45 | 0.62 |
| | SS | 0.00 | 0.00 | 0.00 | -0.40 | 2.56 | -1.81 | -0.39 | 0.00 | 0.06 | 0.04 | -0.10 | 0.00 |
| Sylhet | MK | -2.23* | 0.09 | 1.23 | 1.71 | -0.04 | -0.97 | -0.24 | 0.28 | -0.22 | 0.04 | -0.44 | -0.51 |
| | SS | -0.04* | 0.00 | 0.74 | 2.58 | -0.08 | -1.73 | -0.65 | 0.35 | 0.38 | 0.03 | 0.00 | 0.00 |
| Rajshahi | MK | -1.12 | -0.37 | 0.32 | 0.00 | 1.04 | -1.12 | -2.27* | -0.68 | -0.02 | -0.07 | -0.98 | -0.76 |
| | SS | 0.00 | 0.00 | 0.00 | 0.00 | 0.58 | -1.55 | -3.34* | -0.79 | -0.05 | -0.83 | 0.00 | 0.00 |
| Rangpur | MK | 1.41 | 1.90 | 1.32 | -0.21 | 1.63 | 1.75 | 2.85* | 3.07* | 1.85 | 3.35* | 2.76* | 3.46* |
| | SS | 0.008 | 0.02 | 0.005 | 0.001 | 0.01 | 0.01 | 0.01* | 0.02* | 0.01 | 0.02* | 0.02* | 0.02* |
| Bogra | MK | -1.32 | -0.09 | -2.70* | -0.19 | 1.25 | -0.41 | -0.56 | -0.77 | 1.19 | -0.28 | -1.66 | 0.20 |
| | SS | 0.00 | 0.00 | -0.22* | 0.20 | 0.59 | -0.43 | -0.58 | -0.61 | 1.09 | -0.20 | 0.05 | 0.00 |
| Jessore | MK | 0.58 | 0.47 | -0.13 | -0.37 | 1.39 | 0.37 | 1.63 | -0.52 | 3.18* | 0.43 | 0.21 | 0.77 |
| | SS | 0.00 | 0.00 | 0.00 | -0.03 | 0.72 | 0.39 | 1.10 | -0.48 | 1.58* | 1.09 | 3.30 | 0.00 |
| Khulna | MK | 1.56 | 2.52* | 0.38 | -0.42 | 0.02 | 0.02 | 0.02 | 0.02 | 2.39* | 0.009 | 0.02 | 0.58 |
| | SS | 0.00 | 0.11* | 0.008 | 0.10 | 0.50 | 1.24 | 0.00 | 1.43 | 2.50* | 0.30 | 0.00 | 0.006 |
| Satkhira | MK | 1.44 | 1.20 | 0.63 | 0.74 | 1.74 | -0.19 | 0.57 | -0.19 | 1.63 | -0.12 | 0.12 | 0.28 |
| | SS | 0.00 | 0.02 | 0.03 | 0.25 | 1.04 | -0.18 | 0.40 | 0.11 | 1.17 | 0.07 | 0.00 | 0.00 |
| Barisal | MK | -0.23 | 0.46 | -0.13 | 0.81 | -0.13 | -1.14 | -0.36 | -0.24 | -0.32 | -0.58 | -0.26 | 0.19 |
| | SS | 0.00 | 0.00 | 0.00 | 0.30 | 0.10 | -1.42 | 0.22 | -0.2 | 0.22 | 0.60 | 0.00 | 0.00 |

* Significant trends at 5% significance level. Sen's slope unit is in mm.

**Table 10. Correlation between climate variables and rice yields.**

| Climate variables | Aus | Aman | Boro |
|---|---|---|---|
| Maximum Temperature | 0.51*** | 0.75*** | 0.57*** |
| Minimum Temperature | 0.56*** | 0.42*** | 0.43*** |
| Rainfall | 0.37** | 0.62*** | 0.15 |

*** significance at 1% level.

** significance at 5% level.

**Table 11. Multivariate regression analyses of detrended crop yield.**

| Rice crops | Climate variables | Coefficient | $R^2$ | Adjusted ($R^2$) | F-statistic | p-value of F-statistic |
|---|---|---|---|---|---|---|
| Aus | maxt | -0.19*** | | | | |
| | mint | 0.02 | 0.33 | 0.28 | 6.48 | 0.001 |
| | train | -0.00*** | | | | |
| Aman | maxt | -0.13** | | | | |
| | mint | -0.01 | 0.25 | 0.19 | 4.3 | 0.009 |
| | train | -0.00** | | | | |
| Boro | maxt | -0.04 | | | | |
| | mint | 0.00 | 0.16 | 0.10 | 2.53 | 0.07 |
| | train | -0.00** | | | | |

*** significance at 1% level.

** significance at 5% level.

in *Boro* yield could be explained by climate variability. Moreover, maximum temperature (maxt) and rainfall (train) had negative effects on *Aus* and *Aman* yield, respectively. On the other hand, rainfall had negative effect on *Boro* rice yield.

## Discussion

Detecting the trends of long-term spatial and temporal changes in the temperature and rainfall pattern of Bangladesh and the effects of climate variability on major rice crops was the focus of this research. Bangladesh has experienced an overall significant increasing trend in the annual mean, maximum, minimum temperature, and rainfall. Local variations in temperature and rainfall were also apparent. The country has experienced about a 0.74°C increase in surface warming since 1950 [63]. Between 1949 and 2013, and increased at rates of 0.13°C, 0.13°C, and 0.13°C/decade, respectively. A study [63] found that the average $T_{max}$ and daily average temperature were increased by 1.16°C and 0.47°C, respectively, but the average $T_{min}$ has been decreased by 0.40°C from 1988–2017, which is consistent with our study. Another two studies [4,64] observed similar findings in Bangladesh.

The present study shows that climate change in Bangladesh appears to be consistent with the global expectations of the IPCC report [7]. During the period 1949–2013, the average temperature in Bangladesh increased by about 0.13°C per decade. The fifth assessment from the IPCC reported that between 1951 and 2012, the global mean temperature increased by 0.12°C per decade [7], indicating that Bangladesh has observed more significant warming than identified in the report. The results in this study show more significant warming trends than a previous study [10]. The current study differs from [65,66] in that the former [65] discovered no significant trend in Bangladesh's annual mean, minimum, and maximum temperatures, whereas the latter [66] discovered that Bangladesh's mean annual temperature is increasing at a rate of 0.005°C per year over 100 years. The Met Office [67] reported that Bangladesh had experienced widespread warming since 1960 while also stating that the mean temperature trend during both the summer and winter seasons in Bangladesh had increased at a rate of 0.19°C and 0.24°C per decade, respectively. This analysis was found to be 0.10°C and 0.10°C per decade, respectively, between 1949 and 2013, which is similar to a previous study [67].

The difference in findings between this research and previous studies could be attributed to at least four reasons. First, the previous major studies had been conducted over a 30–50-year period, or the studies did not use a longer time series of data when compared to this research. Secondly, a long-term series of data is included in the sample for this research. Thirdly, while

the previous studies used data from a limited number of stations, the current data is collected from all 35 weather stations in Bangladesh for national scales and 17 selected stations for local scales. Fourthly, non-parametric methods were used for this research to detect trends and changes.

The findings suggest a significant increasing trend in annual rainfall and pre-monsoon rainfall at a rate of 4.20 and 1.35 mm/year, respectively, while other seasons did not show any significant trend at the national level between 1949 and 2013. The findings corroborate with those of [41], where the author observed a significant increasing trend in annual and pre-monsoon rainfall at a rate of +5.5 mm and 2.47 mm/year, respectively, while other seasons did not show any significant trends. The south-western station of Khulna showed a significant increasing trend for annual rainfall, while Comilla and Rajshahi stations showed a significant decreasing trend, and other stations did not observe any significant trend at a 95% confidence level. In addition, regional disparity and variations in annual rainfall studied by [9] that showed upward trends for hilly districts. However. Khulna station, located at the southern coast of Bangladesh, recorded significant rising trends in this present study, which is similar to a previous study [35]. Furthermore, the results are in agreement with [35], where the author suggests a significant increase in pre-monsoon rainfall at Cox's Bazar stations. The increasing rainfall trends in the south and south-eastern Bangladesh might be a result of the rising sea surface temperature (SST) due to global warming [42]. Heavy rainfall occurs in Bangladesh during the monsoons. The main mechanism of rainfall during the summer monsoon in Bangladesh is the south-to-north trajectory monsoon depression in the Bay of Bengal, which turns north-west after being deflected by the Meghalaya Plateau [68]. The declining rainfall in the north-west and west of Bangladesh may be due to the decreased moisture content of these depressions as they move further and further inland [42,68]. A number of studies [44,69–70] projected that monsoon and post-monsoon rainfall in Bangladesh would increase by the end of this century. However, in this research, the south-eastern (Comilla, Maijdee Court), north-eastern (Srimangal), and north-western (Rajshahi) regions of Bangladesh showed a significant decreasing trend for the monsoon region. Furthermore, no stations showed a significant trend for the post-monsoon rainfall. As such, the predictions made by these studies may not be appropriate for the whole region. Additionally, no significant increasing trend for monsoon rainfall was observed at any of the stations either. The reduction of monsoon rainfall in Bangladesh increases the current vulnerability of rain-dependent agriculturists, leading to adverse aggregate impacts across the region.

Temperature and rainfall are the most important weather parameters closely linked to agricultural production. According to the findings of this study, the maximum and minimum temperatures had a significant correlation in the yield of three rice crops: the maximum temperature had the highest strong positive correlation with Aman rice yield and a moderately positive correlation with Aus and Boro rice yield. The minimum temperature had a moderately positive correlation with the yield of three rice crops. On the other hand, rainfall had a positive weak and moderate correlation with *Aus* and *Aman* yields, respectively, and no significant correlation was found for *Boro* rice yield. Moreover, in terms of climate-explained crop variability, maximum temperature and rainfall had a negative effect on *Aus* and *Aman* rice yields, indicating that increases in maximum temperature and rainfall will reduce the yield of the *Aus* and *Aman* rice crop. On the other hand, only rainfall had a negative effect *on Boro* rice yield.

Boro alone contributed 55% of total rice production in Bangladesh [71]. A study [23] predicted that Boro rice production will decline by 20% and 50% for the years 2050 and 2070, respectively, due to the increase in minimum and maximum temperatures. The authors also projected that the daily maximum temperature would exceed 35˚C for the year 2017, and that might cause grain sterility during the growing season. A previous study [72] used various

models to project an 8% decrease in the total rice production in Bangladesh by 2050 due to climate change under a moderate climate scenario. The same study [72] also assumed that irrigation water demand would be 40% to 50% of the dry season water availability in 2050, and this is likely to have a significant negative impact on other water-dependent sectors. Another study [73] has also revealed, through the use of simulated crop models, that the increasing air temperatures will lead to a significant reduction in the yield of Boro rice in Bangladesh by shortening the growing season and reducing the end-of-season total evapotranspiration. Each 1˚C difference in air temperature results in a 4% decrease in total seasonal evapotranspiration, resulting in increased irrigation water demands [73]. A study [73] showed the relationship between climate change and rice yield in Bangladesh using time series data, where the author found that maximum temperature has adverse effects on Boro rice yields, while minimum temperature has a positive and significant effect on Boro rice. However, our research suggests that only rainfall had negative effects on Boro rice yield variability.

Our findings demonstrate how climate affects rice production at various sites and locations. Seasonality and regional climatic variability have an impact on rice crop productivity [20,21]. According to [9,28], the research area's harsh weather conditions, including drought and flooding, make Aman yield the most sensitive to climate change among the three rice crop ecotypes [14]. These results support a study [26] that found a large amount of variation in Aman rice output. The rise in temperature greatly boosted boro yields. The maximum temperature lowered Boro yield, whereas the lowest temperature enhanced it [74]. According to this study, the highest temperature determines whether the lowest temperature has a negative impact on Boro paddy output. A study [37] also found this to be the case. Early on, the Australian paddy needs extra watering [74]. Depending on the area, the pre-monsoon summer season (March to May) contributes for 10–25% of the yearly precipitation [19]. Southwest Bangladesh has severe droughts, and the region's inconsistent rainfall results in long-term environmental catastrophes [64].

In addition, the decreasing monsoon rainfall may also decrease groundwater recharge, which will negatively impact irrigation during the dry season [75]. Rainfall during the dry season is very important for crop cultivation in Bangladesh, in particular Boro rice cultivation. The crop's water demand is only completely met by irrigation processes. Only Khulna station showed a significant trend for winter rainfall, while other stations did not show any significant trends. Bangladesh's north-west and north-central regions support intensive Boro cultivation and exhibit decreasing groundwater trends over the long term [76,77]. Therefore, the declining groundwater will immensely affect rice cultivation in these areas. In addition, the decreasing monsoon rainfall may decrease groundwater recharge, and that will negatively impact dry-season irrigation.

Climate data must be used effectively in climate change adaptation strategies. Better resource management, including the use of climate data, is needed to achieve potential crop yields. The most often used adaptation strategies include altering irrigation, crop variety, tree planting, soil conservation, crop and animal diversity, early and late planting, increasing plant spacing, and adjusting fertilizer treatment quantity and timing [19].

The study's main weakness is that several key factors that affect agricultural yield, such as irrigation, crop variety, soil management, fertilizer usage, and pesticide use, were left out. The datasets used in this study are outdated (1949–2013), which is another drawback. Future studies in Bangladesh should concentrate on forecasting paddy yields in the face of climate change. The results of this research may help regional supervisors and decision-makers establish region-specific adaptation strategies and address food security in Bangladesh by exposing the possible threat connected with climatic fluctuations.

However, the results of this study are expected to provide useful information on the national and local trends of climate change and its effects on major rice yield variability in Bangladesh at a national scale. Our study strongly advises governments and significant sponsors to fund the study, creation, and spread of new rice varieties that are more resilient to drought and other abiotic stressors. Further studies are necessary to find the trends in local and temporal changes in temperature and rainfall and conclude what the large-scale effect of climate change is for Bangladesh.

## Conclusions

This research intends to use multiple statistical methods to explore long-term spatiotemporal climate changes and trends in Bangladesh at the national and regional scales and their implication on rice yield during 1949–2013. The serial correlation and homogeneity tests were performed to check the quality control of the dataset. The results reveal that although Bangladesh is experiencing a significant increasing trend in temperature and rainfall at the national level, but spatial variations in rainfall and temperature at the regional level are more significant than at the national level. The south-eastern, north-eastern, and north-western Bangladesh observed a maximum upward trend for annual $T_{min}$ while south-eastern, north-western, and mid-western Bangladesh observed a maximum upward trend for annual $T_{max}$. Results also show that only May exhibited a significantly increasing trend at a national level. Spatially, south-western Khulna station observed a substantial increasing trend while south-eastern Comilla station observed significant decreasing trends for annual rainfall. Furthermore, the maximum temperature and rainfall had a negative impact on Aus and Aman's rice yield. Rainfall had a negative effect on *Boro's* rice yield.

These findings also support future climate change-induced production reductions. Most regional dummy variables are statistically significant, demonstrating rice output varies by location. Climate change affects different climatic zones differently. Thus, climate change will affect rice production differently in different agroecological zones. Results suggest regional or climate-specific adaption techniques. Site-specific agricultural production and climate change assessments are needed. This research will assist in developing sustainable regional food security policies and action plans. The government may prioritize drought-prone areas when implementing regional food security projects. Our study will enable regional or local adaptation solutions to agricultural output unpredictability, food security, and climate-related rural poverty. The findings of this study will contribute to setting and implementing mitigation and adaptation strategies to meet the challenges of climate change in Bangladesh.

## Acknowledgments

The authors would like to express their sincere gratitude to the Bangladesh Meteorological Department (BMD) for providing temperature and rainfall data for this study and also to Zakaria Hasan for producing maps, and Monica Farrow for improving the language of the manuscript. Logistic support from the Disaster and Development Organisation (DADO) is acknowledged.

## Author Contributions

**Conceptualization:** Edris Alam.

**Data curation:** Al-Ekram Elahee Hridoy, Shekh Md. Shajid Hasan Tusher.

**Formal analysis:** Edris Alam, Al-Ekram Elahee Hridoy, Shekh Md. Shajid Hasan Tusher.

**Funding acquisition:** Edris Alam, Md Kamrul Islam.

**Investigation:** Md Kamrul Islam.

**Methodology:** Al-Ekram Elahee Hridoy, Abu Reza Md. Towfiqul Islam.

**Project administration:** Edris Alam.

**Resources:** Abu Reza Md. Towfiqul Islam, Md Kamrul Islam.

**Software:** Shekh Md. Shajid Hasan Tusher.

**Validation:** Shekh Md. Shajid Hasan Tusher, Abu Reza Md. Towfiqul Islam.

**Visualization:** Shekh Md. Shajid Hasan Tusher, Md Kamrul Islam.

**Writing – original draft:** Edris Alam, Al-Ekram Elahee Hridoy.

**Writing – review & editing:** Abu Reza Md. Towfiqul Islam, Md Kamrul Islam.

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
