## [Decision Letter · Decision Letter 0]

5 Jul 2023

PONE-D-23-16309Climate Change in Bangladesh: Temperature and Rainfall Climatology of Bangladesh for 1949–2013 and its implication on rice yieldPLOS ONE

Dear Dr. Alam,

Thank you for submitting your manuscript to PLOS ONE. After careful consideration, we feel that it has merit but does not fully meet PLOS ONE’s publication criteria as it currently stands. Therefore, we invite you to submit a revised version of the manuscript that addresses the points raised during the review process.

We look forward to receiving your revised manuscript.

Kind regards,

Md. Naimur Rahman

Academic Editor

PLOS ONE

3. We note that Figures 1-3 in your submission contain [map/satellite] images which may be copyrighted. All PLOS content is published under the Creative Commons Attribution License (CC BY 4.0), which means that the manuscript, images, and Supporting Information files will be freely available online, and any third party is permitted to access, download, copy, distribute, and use these materials in any way, even commercially, with proper attribution. For these reasons, we cannot publish previously copyrighted maps or satellite images created using proprietary data, such as Google software (Google Maps, Street View, and Earth). For more information, see our copyright guidelines: http://journals.plos.org/plosone/s/licenses-and-copyright.

a. You may seek permission from the original copyright holder of Figures 1-3 to publish the content specifically under the CC BY 4.0 license. 

Additional Editor Comments:

Major revision requested:

#Reviewer 1

General comments

The study focuses on temperature and rainfall climatology and implication on rice yield. Though the objective of the study is to determine temperature and rainfall climatology implication on rice yield for time series for longer period. The author tried to make the paper comprehensive, but it still requires some major tasks to be completed for improvement of its various sections.

Abstract

1. The abstract is informative up to the mark. It describes fully to reflect the whole study.

Introduction

1. The introduction seems over explanatory; it should be more concise.

2. Line 63: Citation of IPCC 2001 should be updated. I believe IPCC published more exclusive reports regarding the issue after 2001 (ex. Line 65).

3. What is the evidence of this statement in Line 141?

Methodology

1. Overall, analytical techniques are well described in a scientific way and well fitted to the study.

2. Why the data duration is from 1949-2013? The decadal temperature change in results and discussion section. So, did not temperature change in between 2013 to 2023? It is not enough to describe as limitation of this study. It could add more value to this study and most importantly the results could be different. Because, rice yield area (Table-2 of https://journals.plos.org/plosone/article?id=10.1371/journal.pone.0261128), production (Table-3 of https://journals.plos.org/plosone/article?id=10.1371/journal.pone.0261128) has been changed between 2008-2020.

3. It is advised to add data from 2013- till the available date (2023 or 2022) in this study and analyze again to get more specific results.

Results

1. The presentation of results is well organized.

2. According to methodology, results are well described.

3. After adding data from 2013-2023 (2022), please write results accordingly.

Discussion

1. The discussion is also well described.

2. Line 561: Please mention some recent major studies with example. If the studies did not use longer time series data, please show the comparison of the data periods. From line 533-560 author showed comparison of results with other studies, not data sets.

3. After adding data from 2013-20233 (2022), please write discussion accordingly.

Conclusion

1. The revised conclusion should be based on the revised methodology.

Decision

I recommend “Major revision”.

#Reviewer 2

The manuscript has some shortcomings which need to be improved prior to its publication. My recommendation is that the article needs Major Revisions before it can be considered for publication.

1. Abstract: The abstract is a bit generic. Please add some more information regarding your results. It should be improved in a quantitative way.

2. Author should specify the key objectives of this research work in last paragraph of introduction section.

3. Introduction is generalized. I would recommend following recent research articles to reconstruct this section with extensive literature

“Profitable agricultural land use planning in a red and lateritic soil of subtropical environment using field-based index of crop suitability (ICS”

“Field based index of land suitability (ILS): a new approach for rainfed paddy crop production in groundwater scarce region”

4. Methodology section is weakly written. So, my suggestion is to reconstruct it. Author need to review more about RS-GIS environment and current techniques.

“Characterization of groundwater potential zones in water-scarce hardrock regions using data driven model”

“Hydrogeochemical Evaluation of Groundwater Aquifers and Associated Health Hazard Risk Mapping Using Ensemble Data Driven Model in a Water Scares Plateau Region of Eastern India”

“Hydrogeochemical evaluation and corresponding health risk from elevated arsenic and fluoride contamination in recurrent coastal multi-aquifers of eastern India”

“Application of novel data-mining technique based nitrate concentration susceptibility prediction approach for coastal aquifers in India”

“Hydro-chemical assessment of groundwater pollutant and corresponding health risk in the Ganges delta, Indo-Bangladesh region”

Hydrogeochemical characterization based water resources vulnerability assessment in India's first Ramsar site of Chilka lake

5. Discussion section should be written by comparing with already published articles in this concept.

6. In conclusion section, you have to mention the implications of your research and how it makes a footprint in scientific research. Try to incorporate your work to global interest how this research has worldwide importance. It will be interesting for the readers.

7. Reference: Re-check the whole reference just to make sure you have added all the references that you cited in your manuscript.

8. Apart from this the quality of the overall paper is very good. I prefer this article with acceptable with major modifications.

Reviewers' comments:

Reviewer's Responses to Questions

**Comments to the Author**

1. Is the manuscript technically sound, and do the data support the conclusions?

Reviewer #1: Yes

Reviewer #2: Yes

2. Has the statistical analysis been performed appropriately and rigorously? 

Reviewer #1: Yes

Reviewer #2: I Don't Know

3. Have the authors made all data underlying the findings in their manuscript fully available?

Reviewer #1: Yes

Reviewer #2: Yes

4. Is the manuscript presented in an intelligible fashion and written in standard English?

Reviewer #1: Yes

Reviewer #2: Yes

5. Review Comments to the Author

Reviewer #1: General comments

The study focuses on temperature and rainfall climatology and implication on rice yield. Though the objective of the study is to determine temperature and rainfall climatology implication on rice yield for time series for longer period. The author tried to make the paper comprehensive, but it still requires some major tasks to be completed for improvement of its various sections.

Abstract

1. The abstract is informative up to the mark. It describes fully to reflect the whole study.

Introduction

1. The introduction seems over explanatory; it should be more concise.

2. Line 63: Citation of IPCC 2001 should be updated. I believe IPCC published more exclusive reports regarding the issue after 2001 (ex. Line 65).

3. What is the evidence of this statement in Line 141?

Methodology

1. Overall, analytical techniques are well described in a scientific way and well fitted to the study.

2. Why the data duration is from 1949-2013? The decadal temperature change in results and discussion section. So, did not temperature change in between 2013 to 2023? It is not enough to describe as limitation of this study. It could add more value to this study and most importantly the results could be different. Because, rice yield area (Table-2 of https://journals.plos.org/plosone/article?id=10.1371/journal.pone.0261128), production (Table-3 of https://journals.plos.org/plosone/article?id=10.1371/journal.pone.0261128) has been changed between 2008-2020.

3. It is advised to add data from 2013- till the available date (2023 or 2022) in this study and analyze again to get more specific results.

Results

1. The presentation of results is well organized.

2. According to methodology, results are well described.

3. After adding data from 2013-20233 (2022), please write results accordingly.

Discussion

1. The discussion is also well described.

2. Line 561: Please mention some recent major studies with example. If the studies did not use longer time series data, please show the comparison of the data periods. From line 533-560 author showed comparison of results with other studies, not data sets.

3. After adding data from 2013-20233 (2022), please write discussion accordingly.

Conclusion

1. The revised conclusion should be based on the revised methodology.

Decision

I recommend “Major revision”.

Reviewer #2: Manuscript Ref No.: PONE-D-23-16309

The manuscript has some shortcomings which need to be improved prior to its publication. My recommendation is that the article needs Major Revisions before it can be considered for publication.

1. Abstract: The abstract is a bit generic. Please add some more information regarding your results. It should be improved in a quantitative way.

2. Author should specify the key objectives of this research work in last paragraph of introduction section.

3. Introduction is generalized. I would recommend following recent research articles to reconstruct this section with extensive literature

“Profitable agricultural land use planning in a red and lateritic soil of subtropical environment using field-based index of crop suitability (ICS”

“Field based index of land suitability (ILS): a new approach for rainfed paddy crop production in groundwater scarce region”

4. Methodology section is weakly written. So, my suggestion is to reconstruct it. Author need to review more about RS-GIS environment and current techniques.

“Characterization of groundwater potential zones in water-scarce hardrock regions using data driven model”

“Hydrogeochemical Evaluation of Groundwater Aquifers and Associated Health Hazard Risk Mapping Using Ensemble Data Driven Model in a Water Scares Plateau Region of Eastern India”

“Hydrogeochemical evaluation and corresponding health risk from elevated arsenic and fluoride contamination in recurrent coastal multi-aquifers of eastern India”

“Application of novel data-mining technique based nitrate concentration susceptibility prediction approach for coastal aquifers in India”

“Hydro-chemical assessment of groundwater pollutant and corresponding health risk in the Ganges delta, Indo-Bangladesh region”

Hydrogeochemical characterization based water resources vulnerability assessment in India's first Ramsar site of Chilka lake

5. Discussion section should be written by comparing with already published articles in this concept.

6. In conclusion section, you have to mention the implications of your research and how it makes a footprint in scientific research. Try to incorporate your work to global interest how this research has worldwide importance. It will be interesting for the readers.

7. Reference: Re-check the whole reference just to make sure you have added all the references that you cited in your manuscript.

8. Apart from this the quality of the overall paper is very good. I prefer this article with acceptable with major modifications.

6. PLOS authors have the option to publish the peer review history of their article (what does this mean?). If published, this will include your full peer review and any attached files.

Reviewer #1: No

Reviewer #2: No

---

## [Author Response · Author response to Decision Letter 0]

8 Sep 2023

Please find submitted response letter:

On behalf of the authorship of this paper, I would like to thank the Editor and Referees for the review of the original submission and the resubmission that includes a set of very helpful and constructive comments. We have reworked the paper in response to these. A description of the responses to each comment is detailed in the Table below indicating how modifications have been made within the paper. In a very few instances, we have chosen not to follow an individual reviewer suggestion, though provide a detailed clarification where this is the case. I trust this revised manuscript meets all of the standards of the Plos One Journal and am ready to respond to any further points that would arise. 

Yours sincerely,

Dr. Edris Alam

---

## [Decision Letter · Decision Letter 1]

27 Sep 2023

Climate Change in Bangladesh: Temperature and Rainfall Climatology of Bangladesh for 1949–2013 and its implication on rice yield

PONE-D-23-16309R1

Dear Dr. Alam,

We’re pleased to inform you that your manuscript has been judged scientifically suitable for publication and will be formally accepted for publication once it meets all outstanding technical requirements.

Kind regards,

Md. Naimur Rahman

Academic Editor

PLOS ONE

Additional Editor Comments (optional):

I am happy to inform you that your paper has been accepted for publication in PLOS ONE.

Reviewers' comments:

Reviewer's Responses to Questions

**Comments to the Author**

1. If the authors have adequately addressed your comments raised in a previous round of review and you feel that this manuscript is now acceptable for publication, you may indicate that here to bypass the “Comments to the Author” section, enter your conflict of interest statement in the “Confidential to Editor” section, and submit your "Accept" recommendation.

Reviewer #1: All comments have been addressed

Reviewer #2: All comments have been addressed

2. Is the manuscript technically sound, and do the data support the conclusions?

Reviewer #1: Yes

Reviewer #2: Yes

3. Has the statistical analysis been performed appropriately and rigorously? 

Reviewer #1: Yes

Reviewer #2: Yes

4. Have the authors made all data underlying the findings in their manuscript fully available?

Reviewer #1: Yes

Reviewer #2: Yes

5. Is the manuscript presented in an intelligible fashion and written in standard English?

Reviewer #1: Yes

Reviewer #2: Yes

6. Review Comments to the Author

Reviewer #1: Authors addressed all the comments and after review response of the manuscript from authors is up to the mark now to go for printing.

Reviewer #2: The authors have addressed all the review comments accordingly. This manuscript can be accepted in its present form.

7. PLOS authors have the option to publish the peer review history of their article (what does this mean?). If published, this will include your full peer review and any attached files.

Reviewer #1: No

Reviewer #2: No

---

## [Editor Report · Acceptance letter]

3 Oct 2023

PONE-D-23-16309R1 

Climate Change in Bangladesh: Temperature and Rainfall Climatology of Bangladesh for 1949–2013 and its implication on rice yield. 

Dear Dr. Alam:

I'm pleased to inform you that your manuscript has been deemed suitable for publication in PLOS ONE. Congratulations! Your manuscript is now with our production department. 

Kind regards, 

on behalf of

Mr Md. Naimur Rahman 

Academic Editor

PLOS ONE